# The integrated stress response is tumorigenic and constitutes a therapeutic liability in KRAS-driven lung cancer

Nour Ghaddar[1,2], Shuo Wang[1,12], Bethany Woodvine [3,4,12], Jothilatha Krishnamoorthy[1], Vincent van Hoef[5,12], Cedric Darini[1,12], Urszula Kazimierczak[1,6], Nicolas Ah-son[1], Helmuth Popper [7], Myriam Johnson[1,2], Leah Officer [4], Ana Teodósio [4], Massimo Broggini [8], Koren K. Mann[1,9], Maria Hatzoglou [10], Ivan Topisirovic [1,9], Ola Larsson [5], John Le Quesne [3,4,11✉] & Antonis E. Koromilas [1,9✉]

The integrated stress response (ISR) is an essential stress-support pathway increasingly recognized as a determinant of tumorigenesis. Here we demonstrate that ISR is pivotal in lung adenocarcinoma (LUAD) development, the most common histological type of lung cancer and a leading cause of cancer death worldwide. Increased phosphorylation of the translation initiation factor eIF2 (p-eIF2α), the focal point of ISR, is related to invasiveness, increased growth, and poor outcome in 928 LUAD patients. Dissection of ISR mechanisms in *KRAS*-driven lung tumorigenesis in mice demonstrated that p-eIF2α causes the translational repression of dual specificity phosphatase 6 (DUSP6), resulting in increased phosphorylation of the extracellular signal-regulated kinase (p-ERK). Treatments with ISR inhibitors, including a memory-enhancing drug with limited toxicity, provides a suitable therapeutic option for *KRAS*-driven lung cancer insofar as they substantially reduce tumor growth and prolong mouse survival. Our data provide a rationale for the implementation of ISR-based regimens in LUAD treatment.

[1] Lady Davis Institute for Medical Research, Sir Mortimer B. Davis-Jewish General Hospital, Montreal, QC, Canada. [2] Division of Experimental Medicine, Department of Medicine, Faculty of Medicine, McGill University, Montreal, QC, Canada. [3] Leicester Cancer Research Centre, University of Leicester, Leicester, UK. [4] MRC Toxicology Unit, University of Cambridge, Leicester, UK. [5] Department of Oncology-Pathology, Science for Life Laboratory, Karolinska Institute, Solna, Sweden. [6] Department of Cancer Immunology, Chair of Medical Biotechnology, Poznan University of Medical Sciences, Poznan, Poland. [7] Diagnostic and Research Institute of Pathology, Medical University of Graz, Graz, Austria. [8] Laboratory of Molecular Pharmacology IRCCS—Istituto di Ricerche Farmacologiche "Mario Negri", Milan, Italy. [9] Gerald Bronfman Department of Oncology, Faculty of Medicine, McGill University, Montreal, QC, Canada. [10] Department of Genetics, Case Western Reserve University, Cleveland, OH, USA. [11] Beatson Cancer Research Institute, Glasgow, UK. [12] These authors contributed equally: Shuo Wang, Bethany Woodvine, Vincent van Hoef, Cedric Darini. ✉email: John.LeQuesne@glasgow.ac.uk; antonis.koromilas@mcgill.ca

Cancer is a complex collection of genetic diseases characterized by the gain-of-function mutation, amplification, and/or overexpression of key oncogenes together with the loss-of-function mutation, deletion, and/or epigenetic silencing of tumor suppressors[1]. High proliferation rates of cancer cells alter the capacity of the endoplasmic reticulum (ER) machinery in facilitating the folding, assembly, and transport of newly synthesized proteins[2,3]. The protein folding process can be interrupted by cancer-associated forms of stress like DNA damage, proteotoxic, metabolic, and oxidative stress, as a consequence of oncogenic mutations[2,3]. In stressed cells, ER initiates a group of signal transduction pathways, collectively termed the unfolded protein response (UPR), whose primary role is to restore homeostasis through increased endurance and adaptation[2,3]. When disruption of ER homeostasis is extreme and prolonged, UPR activates pro-death programs to protect the host from the deleterious effects of damaged cells[2,3].

The ER-resident kinase PERK (*EIF2AK3*) is a component of UPR and member of the integrated stress response (ISR) for the regulation of mRNA translation under stress[4]. In addition to PERK, ISR consists of three other kinases, namely, the heme-regulated kinase HRI, the general control nonderepressible GCN2, and the RNA-activated protein kinase PKR, which exhibit high specificity for the phosphorylation of the alpha (α) subunit of the eukaryotic translation initiation factor 2 at serine 52 (p-eIF2α) in response to distinct forms of stress[4]. eIF2 is a trimeric complex of α, β and γ subunits, which is recruited to 40S ribosomal subunits as a ternary complex (TC) with the initiator Met-tRNA$_i$^Met and GTP[5]. Upon initiation codon recognition, GTP is hydrolyzed and eIF2-GDP is released from the ribosome. The guanine nucleotide exchange factor (GEF) eIF2B is required to exchange GDP for GTP to enable the participation of eIF2-GTP in subsequent rounds of initiation[6]. In response to oncogenic or therapy-induced environmental stressors, p-eIF2α exhibits a high affinity for binding to the limiting amounts of eIF2B, leading to sequestration of the latter and reduced eIF2-GDP to eIF2-GTP recycling[6]. Decreased TC availability hinders global translation, while permitting translation of a subset of mRNAs encoding transcriptional and cell fate regulators of the stress response[7]. PERK and GCN2 form the pro-survival and adaptive arms of ISR as opposed to PKR and HRI, which are mainly pro-apoptotic[8–10].

While ISR is upregulated in many cancers[8], its precise function in tissue-specific cancer initiation and progression remains poorly understood. Here, we examined the implication of ISR in the development of lung adenocarcinoma (LUAD), which constitutes 40% of all lung malignancies[11]. At the time of diagnosis, most LUAD patients have already developed an advanced disease and the median survival barely exceeds 18 months from diagnosis[12]. We show that upregulation of p-eIF2α, which is the hallmark of ISR, substantiates a significant decrease in the probability of overall, cancer-specific, and recurrence-free survival of 928 patients by ~12 months. Considering that ~30% of LUAD cases are attributed to the activating mutations of Ki-ras2 Kirsten rat sarcoma viral oncogene homolog (*KRAS*)[13], we investigated the role of ISR in *KRAS*-driven lung carcinogenesis. Our findings demonstrate the tumorigenic function of the PERK/p-eIF2α arm of ISR together with the strong therapeutic benefits of its pharmacological inhibition for the treatment of mutant *KRAS* lung cancer. Because mutant *KRAS* cancers are largely refractory to therapy[14], our data show that the adaptive role of ISR in the addiction of tumors to *KRAS* mutations is a rational target for the implementation of effective therapies against a deadly form of lung cancer.

## Results

**Increased p-eIF2α is related to poor outcome in human LUADs.** Adopting tissue microarrays (TMAs) derived from a continuous cohort of 928 primary human LUADs we obtained data showing that patients positive for p-eIF2α have a significantly poorer outcome compared to patients negative for p-eIF2α (Fig. 1a, b; Supplementary Fig. 1). Importantly, we identify an association between p-eIF2α and tumor growth pattern, WHO tumor type, and tumor cell proliferation with higher p-eIF2α levels in regions with invasive patterns compared to in situ growth, and highest of all in regions with the highly aggressive solid growth pattern (Fig. 1c–e; Table 1). Early in situ/minimally invasive lesions (AIS/MIA) had the lowest levels of p-eIF2α, followed by predominantly in situ tumors showing clear areas of invasion (lepidic-predominant adenocarcinoma), and then the invasive-predominant groups (Fig. 1d). Again, solid-predominant tumors had the highest p-eIF2α levels. Interestingly, mucinous tumors, which form a biologically distinct subgroup of primary LUAD samples[15], showed very low p-eIF2α levels. The association between p-eIF2α and tumor cell proliferation (Ki67 positivity) showed a significant positive relationship between p-eIF2α and tumor cell proliferation (Spearman's Rho 0.361, $p < 2.2 \times 10^{-16}$) (Fig. 1e). These data identified p-eIF2α as a prognostic marker in human LUAD and provided a role for signaling through the ISR in driving both tumor cell proliferation and invasion.

**p-eIF2α promotes mutant KRAS-driven lung tumor formation in mice.** To investigate the role of ISR in lung cancer, we employed KRAS^+/LSL-KRAS G12D mice bearing a loxP-STOP-LoxP (LSL)-*KRAS G12D* allele, which is conditionally activated in the lungs by viral vectors expressing CRE recombinase[16]. KRAS^+/LSL-KRAS G12D mice were crossed with mice containing either a conditional homozygous S51A mutation of *eIF2S1* allele (fTg/0; eIF2α^A/A) or wild type *eIF2S1* (fTg/0;eIF2α^S/S) (Fig. 2a)[17]. Lung tumor formation in the offspring mice was induced by infection with lentiviruses expressing CRE under the control of carbonic anhydrase 2 promoter[18], which is active in type I and II alveolar epithelial lung cells[19]. The CRE lentiviruses also produced TP53 shRNA from an U6/H1 promoter to accelerate lung tumor formation[18]. Mice with KRAS G12D eIF2α^A/A tumors survived ~18 weeks longer than mice with KRAS G12D eIF2α^S/S tumors (Fig. 2b). Ultrasound imaging of live mice showed a higher number of lung tumors in eIF2α^S/S than eIF2α^A/A mice as early as 7 weeks after conditional KRAS G12D expression (Fig. 2c). There was a substantial increase in the size of eIF2α^S/S lung tumors when compared to eIF2α^A/A tumors at 18 and 28 weeks after KRAS G12D induction (Fig. 2c). Hematoxylin and eosin (H&E) staining of the lungs revealed the presence of adenocarcinoma in situ, carcinomas, and invasive adenocarcinoma lesions in both KRAS G12D eIF2α^S/S and eIF2α^A/A mice[20]; these lesions were larger in eIF2α^S/S than eIF2α^A/A lungs (Fig. 2d).

Transplantation of primary KRAS G12D eIF2α^S/S and eIF2α^A/A lung tumor cells in immune-deficient (nu/nu) mice supported a cell-autonomous tumorigenic function of p-eIF2α (Fig. 2e). Because p-eIF2α impacts on gene transcription[21,22], we performed RNA-seq of KRAS G12D eIF2α^S/S and eIF2α^A/A tumor cells to identify genes and pathways potentially linked to observed differences in tumor phenotypes. Data analyses identified 2249 genes that were differentially regulated in eIF2α^S/S vs. eIF2α^A/A tumor cells (Fig. 2f). Specifically, 983 genes were decreased, and 1266 genes were increased in eIF2α^S/S vs. eIF2α^A/A cells (Fig. 2f). We performed Upstream Regulator (UR) analysis as implemented in Ingenuity® Pathway Analysis (IPA) to identify upstream regulators driving p-eIF2α dependent transcriptional reprogramming. UR analysis predicted the activation of the eIF2α kinase PERK (*EIF2AK3*) and transcription factors *ATF4*, *DDIT3* (*CHOP*) and CREB-binding protein (*CREBBP*), which are well-established ISR constituents and UPR effectors (Fig. 2g)[21–23]. In addition, UR

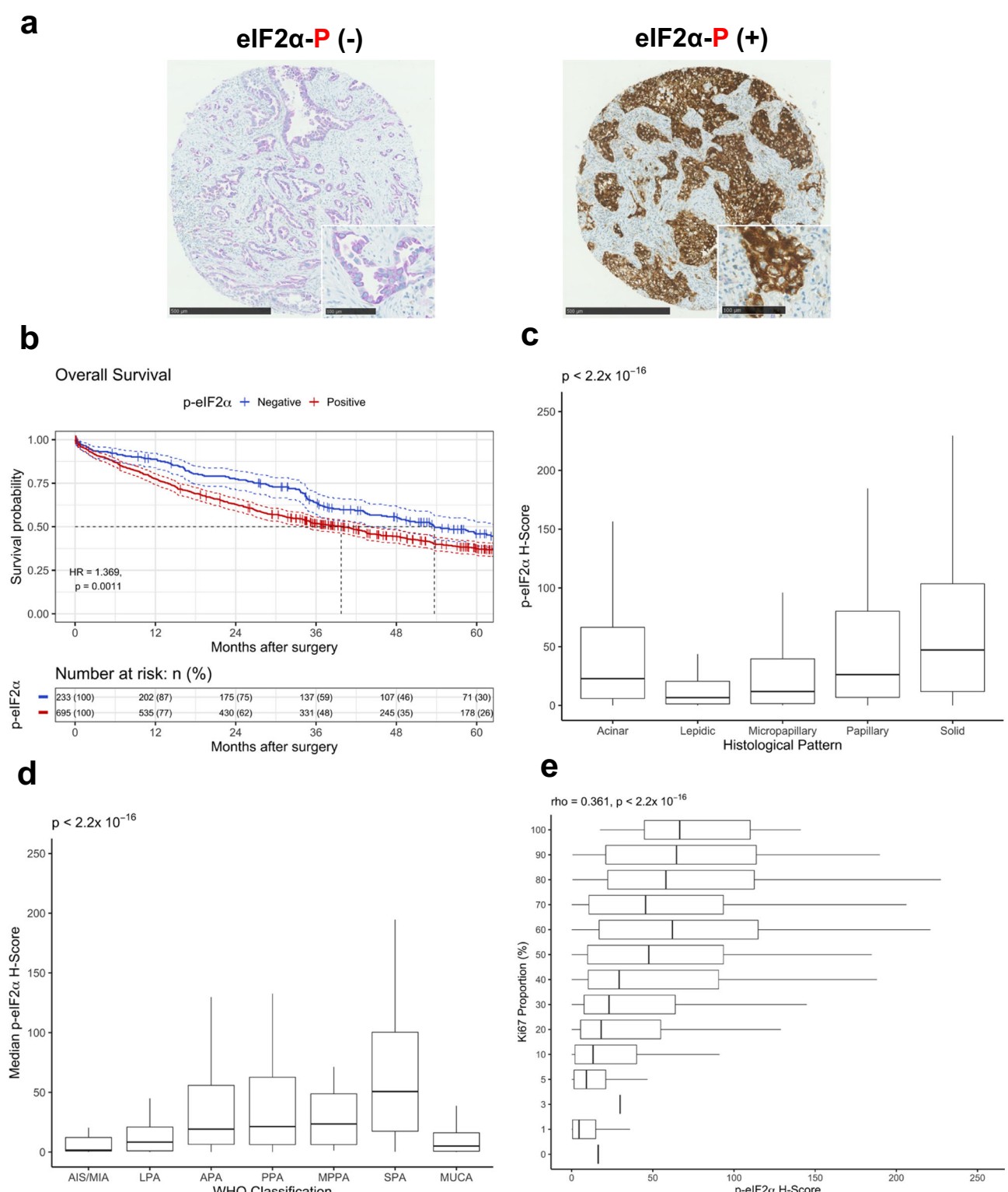

analysis indicated the upregulation of ERK signaling by p-eIF2α, which plays a major role in *KRAS*-driven cancer growth[24]. In line with this prediction, immunohistochemical (IHC) analysis revealed higher levels of p-ERK in KRAS G12D eIF2α$^{S/S}$ than eIF2α$^{A/A}$ lung tumors regardless of the histological tumor type (Fig. 3a; Supplementary Fig. 2). The direct relationship between p-eIF2α and p-ERK also became evident in a different mouse lung cancer model, which was driven by *KRAS Q61R/L* after urethane (ethyl carbamate) treatment[25]. Specifically, lung tumors in

urethane-treated mice, which carried a heterozygous germline S51A substitution of *eIF2S1* (eIF2α$^{S/A}$)[26], were fewer and smaller than wild type eIF2α$^{S/S}$ mice and characterized by decreased proliferation (Ki67) and p-ERK compared to eIF2α$^{S/S}$ lung tumors (Supplementary Fig. 3). As with mouse lung tumors, multiplex fluorescent IHC analyses of LUAD patient specimens revealed significant positive relationships at a single-epithelial cell level between cytoplasmic p-eIF2α and nuclear p-ERK (Supplementary Fig. 4).

**Fig. 1 p-eIF2α prognosticates the poorer survival of 928 patients with primary LUADs after surgery. a** Representative images of human LUADs designated as "negative" (−) or "positive" (+) for p-eIF2α. 2411 tumor cores were stained; 656 were negative and 1755 were positive for p-eIF2α. Scale bars correspond to 500 and 100 μm on core image and enlarged image, respectively. **b** Kaplan–Meier survival curves of p-eIF2α expression for patients' overall survival after surgery (blue curve associates with patients negative (−) for p-eIF2α and red curve associates with patients positive (+) for p-eIF2α). Significance was determined using log-rank test (two-sided). Confidence intervals are represented by the dashed lines around the survival curves. HR = hazard ratio. **c** p-eIF2α H-scores plotted against core/regional growth pattern (lepidic $n = 286$, acinar $n = 917$, papillary $n = 266$, micropapillary $n = 85$, solid $n = 636$). (Kruskal–Wallis chi-squared (two-sided) $= 197.21$, $p < 2.2 \times 10^{-16}$, excludes outlying values). **d** Median p-eIF2α H-scores plotted against WHO tumor type (adenocarcinoma in situ/minimally invasive adenocarcinoma (AIS/MIA) $n = 22$, lepidic-predominant adenocarcinoma (LPA) $n = 87$, acinar-predominant adenocarcinoma (APA) $n = 330$, papillary-predominant adenocarcinoma (PPA) $n = 131$, micropapillary-predominant adenocarcinoma (MPPA) $n = 20$, solid-predominant adenocarcinoma (SPA) $n = 254$, mucinous adenocarcinoma (MUCA) $n = 65$) (Kruskal–Wallis chi-squared (two-sided) $= 144.81$, $p < 2.2 \times 10^{-16}$, excludes outlying values). **e** p-eIF2α H-score plotted against core/regional Ki67 proportion within tumor tissue (Spearman's Rho $= 0.361$, $p < 2.2 \times 10^{-16}$, excludes outlying values). For box plots (**c–e**), the three solid lines represent the 75% percentile, the median, and the 25% percentile in turn. The whisker boundaries represent ±1.5 * IQR, (IQR = 75% percentile – 25% percentile).

**Table 1 Univariate and multivariate overall survival model including phospho-eIF2α (eIF2α-P) H-score, stage, sex, performance status, and WHO classification. Only complete cases included [$n = 842$, number of events = 583, number of patient years = 18 (1998–2015)].**

| | | Univariate Cox Model | | | Multivariate Cox Model | | |
|---|---|---|---|---|---|---|---|
| | | HR | 95% CI | *p*-value | HR | 95% CI | *p*-value |
| eIF2α-P (Negative vs Positive) | Positive | 1.427 | 1.162–1.752 | 0.00068 | 1.091 | 0.882–1.349 | 0.422 |
| Stage (1 vs 2 vs 3+) | 2 | 1.128 | 0.786–1.619 | 0.510 | 1.086 | 0.756–1.561 | 0.655 |
| | 3 | 2.059 | 1.579–2.686 | $9.95 \times 10^{-8}$ | 1.828 | 1.395–2.395 | $1.21 \times 10^{-5}$ |
| Sex (Male vs Female) | Female | 0.676 | 0.574–0.795 | $2.42 \times 10^{-6}$ | 0.719 | 0.609–0.849 | $9.77 \times 10^{-5}$ |
| Performance Status (0 vs 1 vs 2+) | 1 | 1.269 | 1.047–1.537 | 0.0156 | 1.284 | 1.059–1.556 | 0.0110 |
| | 2 | 2.012 | 1.638–2.470 | $2.54 \times 10^{-11}$ | 1.986 | 1.616–2.441 | $6.83 \times 10^{-11}$ |
| WHO (AIS/MIA/LPA vs APA/PPA vs SPA/MPPA) | APA/PPA | 1.932 | 1.418–2.631 | $2.93 \times 10^{-5}$ | 1.682 | 1.229–2.301 | 0.00115 |
| | SPA/MPPA | 2.348 | 1.705–3.234 | $1.73 \times 10^{-7}$ | 1.786 | 1.283–2.486 | 0.000586 |

**p-eIF2α stimulates p-ERK via the translational repression of DUSP6.** To determine the mechanism of increased p-ERK by p-eIF2α, we analyzed the regulation of the mitogen-activated protein kinase (MAPK) pathway in mouse KRAS G12D eIF2α$^{S/S}$ and eIF2α$^{A/A}$ lung tumor cells. While p-ERK levels were substantially lower in KRAS G12D eIF2α$^{A/A}$ than eIF2α$^{S/S}$ cells, the amount of active p-MEK, which phosphorylates ERK, did not differ between the two cell types (Fig. 3b). This result was conspicuous of a role of p-eIF2α in the regulation of dual-specificity phosphatases (DUSPs), which act downstream of MEK to inactivate ERK through the dephosphorylation of TxY motif[27]. Loss of p-eIF2α in KRAS G12D lung tumor cells was accompanied by an upregulation of DUSP6 (Fig. 3b), which is highly selective in its ability to dephosphorylate ERK among the DUSP family members[27]. ATF4 expression, which is upregulated by p-eIF2α at the translational level[7], was substantially increased in KRAS G12D eIF2α$^{S/S}$ compared to eIF2α$^{A/A}$ cells, supporting the translation function of p-eIF2α in mouse KRAS G12D cells (Fig. 3b). Polysome profiling of KRAS G12D tumor cells indicated that p-eIF2α inhibits DUSP6 at the translational level. Specifically, analyses of total and poly-ribosomal RNA revealed a substantial enrichment of DUSP6 mRNA specifically in the poly-ribosomal fraction of eIF2α$^{A/A}$ compared to eIF2α$^{S/S}$ tumor cells after normalization to glyceraldehyde 3-phosphate dehydrogenase (GAPDH) and ACTIN mRNAs in the corresponding total and poly-ribosomal mRNA fractions (Fig. 3c). Moreover, IHC analysis of mouse lung sections indicated that loss of p-eIF2α was associated with increased cytoplasmic DUSP6 expression in KRAS G12D tumors (Fig. 3d).

DUSP6 downregulation accounted for the upregulation of p-ERK by p-eIF2α because treatment KRAS G12D eIF2α$^{A/A}$ cells with a mix of 4 different DUSP6 siRNAs restored p-ERK to equal levels of p-ERK in KRAS G12D eIF2α$^{S/S}$ cells treated with scrambled siRNAs (Fig. 3e). Moreover, DUSP6 downregulation by siRNAs increased the survival of KRAS G12D eIF2α$^{A/A}$ cells as

much as the survival of KRAS G12D eIF2α$^{S/S}$ under the same treatment (Fig. 3f). Furthermore, treatment with BCI, a small-molecule DUSP6 inhibitor[28,29], restored the differences in p-ERK between KRAS G12D eIF2α$^{S/S}$ and eIF2α$^{A/A}$ cells (Supplementary Fig. 5). Thus, translational suppression of DUSP6 by p-eIF2α justifies the increased p-ERK and survival of mutant KRAS lung tumors.

**Mutant KRAS upregulates the PERK/p-eIF2α arm in human LUAD cells.** Considering the implication of PERK in mouse KRAS G12D tumorigenesis from the UR analysis of the RNA-seq data (Fig. 2g), we investigated the connection between mutant KRAS and PERK/p-eIF2α arm in human LUAD cells. We noticed an increase in p-PERK and p-eIF2α in H23 and H358 cells containing KRAS G12C compared to H1703 cells containing wild type (WT) KRAS (Supplementary Fig. 6a). Upregulation of PERK/p-eIF2α arm was associated with increased p-ERK in KRAS G12C cells compared to WT KRAS cells and was further enhanced by treatment with the ER stressor thapsigargin (TG) in the mutant KRAS cells (Supplementary Fig. 6a). Also, ectopic expression of KRAS G12C, G12D or G12V in H1299 cells containing WT *KRAS* alleles resulted in the upregulation of PERK/p-eIF2α arm along with increased p-ERK compared to WT KRAS-overexpressing H1299 cells (Supplementary Fig. 6b). Because H1299 cells harbor a *NRAS* Q61K allele[30], which could interfere with the effects of mutant KRAS overexpression, we tested the effects of KRAS G12C overexpression in H1703 lung cancer cells containing WT alleles of all the *RAS* isoforms. As with H1299 cells, KRAS G12C overexpression in H1703 cells stimulated the p-PERK, p-eIF2α, and p-ERK compared to isogenic cells over-expressing WT KRAS (Supplementary Fig. 6c). Thus, mutant KRAS upregulates the PERK/p-eIF2α arm of ISR in human LUAD cells. We further tested the implication of PERK/p-eIF2α

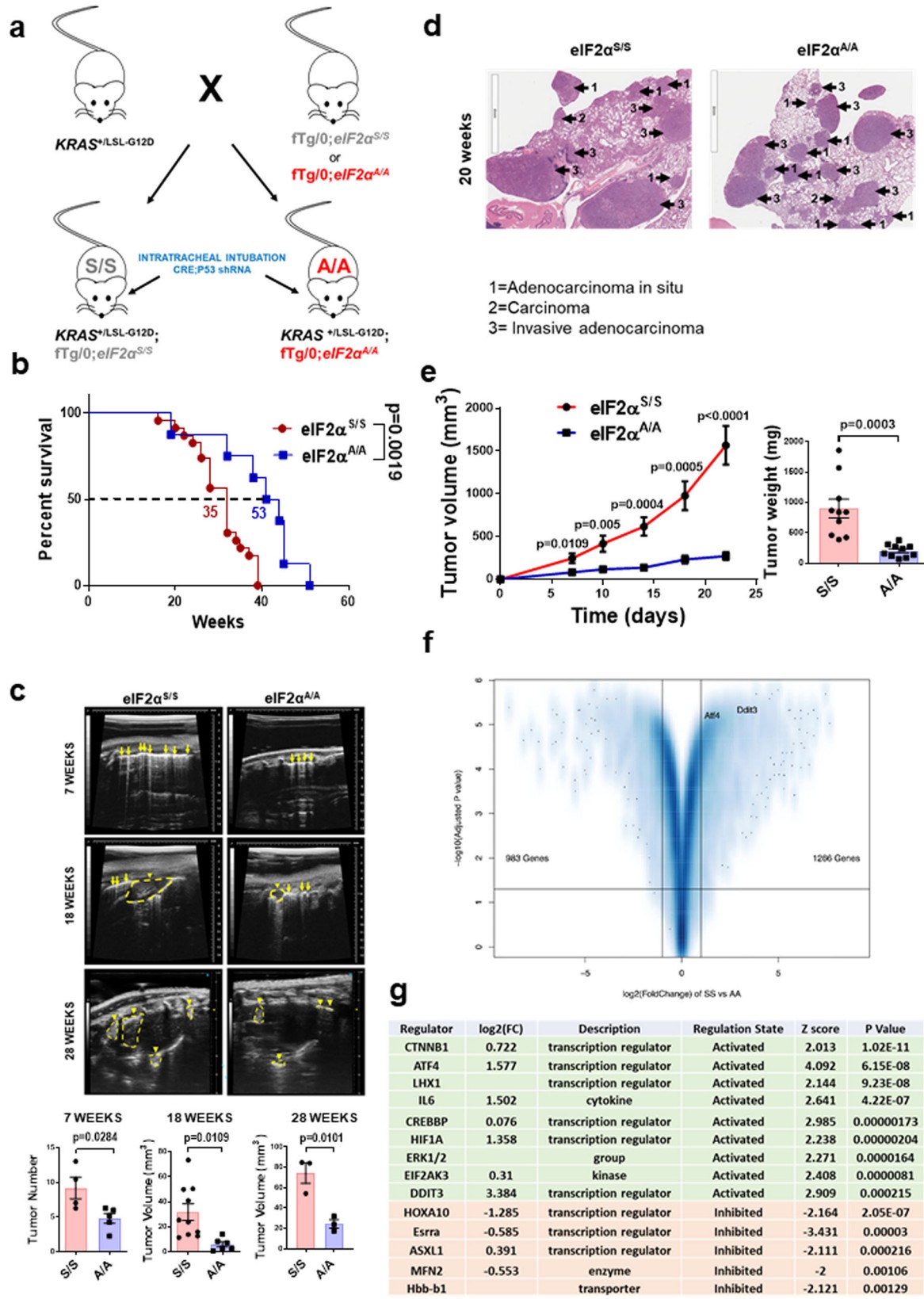

arm in the survival of human LUAD cells. PERK inhibition by GSK2606414, a potent and selective PERK inhibitor among the eIF2α kinases[31], caused a stronger reduction in the colony-forming efficacy of human LUAD cells with KRAS G12C than WT KRAS (Supplementary Fig. 7a). The increased susceptibility to PERK inhibition was also evident for human lung cancer cells overexpressing KRAS G12C compared to their isogenic cells overexpressing WT KRAS (Supplementary Fig. 7b). Thus, upregulation of PERK/p-eIF2α arm by mutant KRAS renders human LUADs increasingly susceptible to PERK inhibition.

**Fig. 2 p-eIF2α promotes KRAS G12D lung tumorigenesis in mice. a** Mouse mating scheme. KRAS$^{+/\text{LSL-G12D}}$ mice containing a latent CRE-loxP KRAS G12D allele were crossed with either fTg/0;eIF2α$^{S/S}$ or fTg/0;eIF2α$^{A/A}$ mice. The offspring mice were subjected to intra-tracheal intubation of CRE-expressing lentiviruses and TP53 shRNA. **b** Survival curve of KRAS G12D eIF2α$^{S/S}$ (red curve, $n = 23$) and eIF2α$^{A/A}$ (blue curve, $n = 8$) mice subjected to intra-tracheal intubation with $6 \times 10^5$ functional lentiviral particles per mouse. Data represent Mean ± SEM with two-sided Log-rank, Mantel-Cox Test, $P$ value = 0.0019. **c** Detection of mouse lung tumors in the septum, located peripherally in contact with the pleura, by Ultrasound imaging at 7 weeks (eIF2α$^{S/S}$ $n = 4$, eIF2α$^{A/A}$ $n = 5$, $P$ value = 0.028), 18 weeks (eIF2α$^{S/S}$ $n = 10$, eIF2α$^{A/A}$ $n = 6$, $P$ value = 0.0109) and 28 weeks of KRAS G12D expression (eIF2α$^{S/S}$ $n = 3$, eIF2α$^{A/A}$ $n = 3$, $P$ value = 0.0101). Tumor location is indicated by an arrow and tumor size by intermittent lines in yellow color. **d** Representative H&E staining of KRAS G12D eIF2α$^{S/S}$ ($n = 5$) and eIF2α$^{A/A}$ ($n = 5$) lung sections at 20 weeks post-induction of KRAS G12D. Arrows indicate types of tumors. 1 = adenocarcinoma in situ. 2 = Carcinoma. 3 = invasive adenocarcinoma. Scale bars correspond to 4 mm of core images. **e** Subcutaneous growth of primary KRAS G12D eIF2α$^{S/S}$ and eIF2α$^{A/A}$ lung tumors in nude mice ($n = 10$, $P$ value = 0.0003). Tumor mass (mg) at the endpoint of the experiment is shown in the histogram graph. (**f**) Volcano plot visualizing the differential mRNA expression between eIF2α$^{S/S}$ and eIF2α$^{A/A}$ mouse KRAS G12D tumors. **g** Top activated or inhibited inferred upstream regulators according to the Upstream Regulator (UR) analysis as implemented in IPA comparing eIF2α$^{S/S}$ vs eIF2α$^{A/A}$ (**c**, **e**) Data represent Mean ± SEM. Significance in differences between two datasets was determined using two-tailed unpaired $t$-test. $P$ values are indicated on the graphs.

**PERK/p-eIF2α arm suppresses DUSP6 and increases p-ERK in mutant KRAS lung tumors**. We next addressed the implication of PERK in p-eIF2α and the survival of mouse KRAS G12D cells. Treatment with PERK siRNAs impaired PERK autophosphorylation at T980 (p-PERK) and decreased p-eIF2α in KRAS G12D eIF2α$^{S/S}$ cells prior to and after exposure to ER stressor TG (Fig. 4a). PERK downregulation by siRNAS reduced the survival of KRAS G12D eIF2α$^{S/S}$ but not of eIF2α$^{A/A}$ cells in colony formation assays (Fig. 4b). Decreased survival of KRAS G12D eIF2α$^{S/S}$ cells by the loss of PERK was equal to the survival of eIF2α$^{A/A}$ cells subjected to scrambled siRNA treatments (Fig. 4b). The modest increase of KRAS G12D eIF2α$^{A/A}$ cell survival by PERK downregulation could be attributed to p-eIF2α-independent PERK functions as previously reported[8] (Fig. 4b). As with siRNAs, treatment with the PERK inhibitor GSK2606414 substantially decreased the survival of KRAS G12D eIF2α$^{S/S}$ but not of eIF2α$^{A/A}$ cells (Fig. 4c). PERK inhibition by GSK2606414 promoted the death of KRAS G12D eIF2α$^{S/S}$ but not of eIF2α$^{A/A}$ cells as determined by flow cytometry analysis (Supplementary Fig. 8). Collectively, the data supported the reliance of mouse KRAS G12D lung tumor cells on the PERK/p-eIF2α arm for survival.

We further assessed the role of PERK in the regulation of p-ERK and DUSP6 in mutant KRAS lung tumor cells, which were treated with the ER stress inducer TG to attain robust stimulation of PERK. While TG treatment-induced p-PERK in mouse KRAS G12D eIF2α$^{S/S}$ and eIF2α$^{A/A}$ cells, activated PERK was associated with increased p-ERK and decreased DUSP6 expression exclusively in eIF2α$^{S/S}$ cells (Fig. 4d). Successive treatments of TG-stressed cells with increasing concentrations of the PERK inhibitor GSK2606414 gradually reduced p-eIF2α, p-ERK, and increased DUSP6 expression in KRAS G12D eIF2α$^{S/S}$ cells only (Fig. 4d). As with mouse KRAS G12D cells, stimulation of PERK autophosphorylation at T982 (p-PERK) by TG treatment was associated with increased p-eIF2α and p-ERK as well as decreased DUSP6 expression in human lung cancer cells with KRAS G12C (H23, H358) but not in cells with WT KRAS (H1299, H1703) (Supplementary Fig. 9). Subsequent treatments of TG-stressed cells with increasing concentrations of the PERK inhibitor GSK2606414 decreased p-eIF2α and p-ERK but increased DUSP6 expression in mutant KRAS cells only (Supplementary Fig. 9). These data suggested that stimulation of the PERK/p-eIF2α branch results in the downregulation of DUSP6 and upregulation of p-ERK specifically in mutant KRAS lung cancer cells.

We further aimed at disrupting p-eIF2α function in mutant KRAS lung tumor cells by treatments with the ISR inhibitor (ISRIB), a small molecule antagonist of the translational effects of p-eIF2α[32]. Increasing amounts of ISRIB prevented the inhibition of DUSP6 expression and stimulation of p-ERK in KRAS G12D eIF2α$^{S/S}$ cells in response to TG treatment as opposed to eIF2α$^{A/A}$ cells, which remained unresponsive to treatments (Fig. 5a). The efficacy of ISRIB to antagonize the translational effects of p-eIF2α was verified by the inhibition of ATF4 expression in TG-treated KRAS G12D eIF2α$^{S/S}$ cells only (Fig. 5a). These findings supported the specificity of ISRIB in the inhibition of p-ERK by increased p-eIF2α in mutant KRAS G12D cells. In human lung cancer cells, ISRIB acted in a dose-dependent manner to increase DUSP6 and decrease p-ERK in TG-treated H23 and H358 cells with KRAS G12C but not in H1299 and H1703 cells with WT KRAS (Fig. 5b). These results supported the interpretation that inhibition of the translational function of p-eIF2α by ISRIB is an efficient means to impair p-ERK in mouse and human lung tumor cells with *KRAS* mutations.

**Pharmacological disruption of PERK/p-eIF2α arm impairs mutant KRAS lung cancer formation**. The tumorigenic function of PERK/p-eIF2α arm prompted us to investigate the therapeutic potential of its pharmacological inhibition in the treatment of *KRAS*-driven lung cancer. To this end, we examined the growth of H1299 cells overexpressing either WT KRAS or KRAS G12C in immune-deficient mice treated with the PERK inhibitor GSK2606414 or ISRIB. Treatments of mice with each inhibitor caused a significant reduction in the growth of KRAS G12C cells as opposed to isogenic WT KRAS cells, which were less responsive to treatments especially with ISRIB (Fig. 6a, b).

We next tested the effects of ISR inhibitors on the growth of mouse Lewis Lung Carcinoma (LLC) cells containing a *KRAS* G12C allele[33] in orthotopic transplantation assays in immuno-competent mice. Treatments were initiated on the 12th day after the intratracheal intubation of the LLC cells, at which point mice had developed detectable tumors by H&E staining and ultrasound imaging of the lungs (Supplementary Fig. 10a). Analysis of tumor growth in mice by ultrasound imaging of the lungs indicated a significant reduction in LLC growth between the 3rd and 6th week of treatment with either the PERK inhibitor GSK2606414 or ISRIB compared to mice treated with vehicle control (Fig. 6c). IHC analysis of lung sections indicated that ISRIB increased DUSP6 and decreased p-ERK in tumors along with ATF4 downregulation, which served as a marker of ISRIB treatment (Fig. 6d). Tumor inhibition was associated with decreased proliferation (i.e., Ki-67) and increased apoptosis (i.e., cleaved Caspase 3) indicative of the anti-tumor effects of the treatments (Supplementary Fig. 10b).

We further assessed the effects of ISR inhibition on mouse KRAS G12D tumor growth in syngeneic mice. Treatment of mice bearing sizable (~200 mm³) subcutaneous KRAS G12D tumors with ISRIB resulted in a substantial suppression of tumor growth (Supplementary Fig. 11a). Consequently, we tested the effects of ISRIB on the growth of tumors formed by the conditional

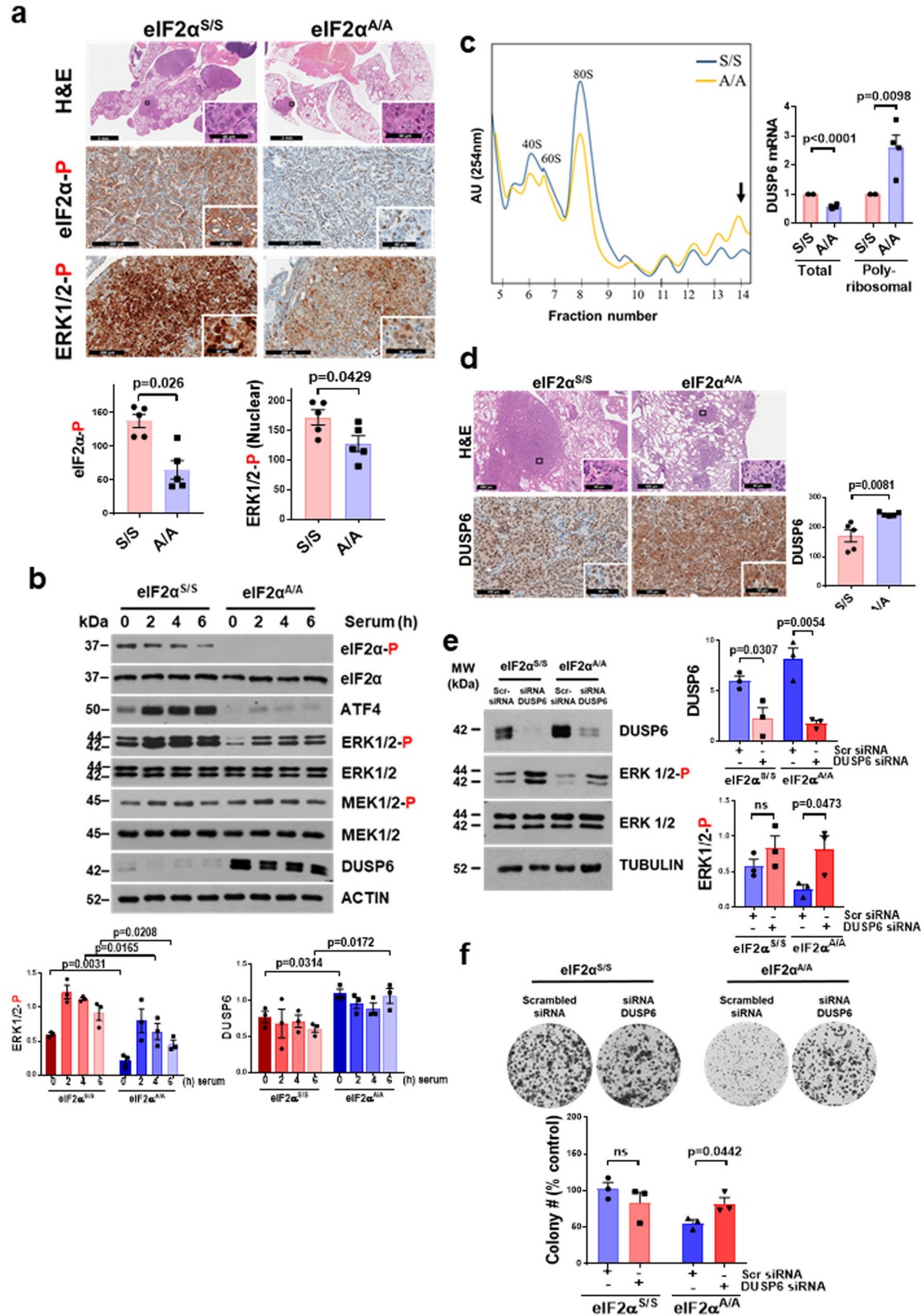

activation of KRAS G12D in the mouse lungs. ISRIB treatment of mice was initiated 10 weeks after the intratracheal intubation of CRE lentivirus, at which point KRAS G12D tumors were readily detected by ultrasound imaging of the lungs (Supplementary Fig. 11b). Monitoring tumor development by ultrasound imaging, we noticed that ISRIB treatments caused a substantial reduction

in lung tumor size after 24 weeks of treatment (Fig. 6e). The difference in tumor growth between vehicle control- and ISRIB-treated mice was maintained at 38 weeks (Fig. 6e), at which time ~50% of vehicle-treated mice were deceased (Fig. 6f). The life span of ISRIB-treated mice was significantly prolonged compared to vehicle control-treated mice (Fig. 6f). Specifically, all vehicle

**Fig. 3 p-eIF2α stimulates p-ERK through the translational repression of DUSP6. a** H&E staining of mouse lungs and IHC staining of tumors for phospho-eIF2 (eIF2α-P) and phospho-ERK (ERK-P) at 20 weeks after CRE-lentivirus intubation (eIF2α$^{S/S}$ $n = 5$, eIF2α$^{A/A}$ $n = 5$ $P$ value = 0.0026). Graphs represent the average H-score of tumors per lung section of eIF2α$^{S/S}$ and eIF2α$^{A/A}$ mice. Scale bars for H&E staining correspond to 3 and 2 mm of the core images of eIF2α$^{S/S}$ and eIF2α$^{A/A}$ tumors, respectively, and 60 μm of enlarged images. For IHC staining images, scale bars correspond to 200 and 60 μm of core and enlarged tumor images, respectively. **b** KRAS G12D eIF2α$^{S/S}$ and eIF2α$^{A/A}$ tumor cells were deprived from serum for 18 h followed by serum stimulation (10% FBS) for the indicated time. Protein extracts (50 μg) were subjected to immunoblotting for the indicated proteins. The graphs represent quantitative analyses from three biological replicates. Phosphorylated ERK (ERK-P) was normalized to total ERK and DUSP6 to either ACTIN or TUBULIN. **c** Polysome profiling of KRAS G12D eIF2α$^{S/S}$ and eIF2α$^{A/A}$ cells. Arrowhead indicates the poly-ribosomal fraction used for mRNAs detection. DUSP6 mRNA was normalized to glyceraldehyde 3-phosphate dehydrogenase (GAPDH) and ACTIN mRNAs in total ($n = 4$, $P$ value < 0.0001) and poly-ribosomal fractions ($n = 4$, $P$ value = 0.0098). **d** H&E staining of mouse lungs and IHC staining of tumors for DUSP6 at 20 weeks after KRAS G12D induction. Graphs represent the average H-score of tumors per lung section of eIF2α$^{S/S}$ ($n = 5$) and eIF2α$^{A/A}$ ($n = 5$) mice with $P$ value = 0.0081. Scale bars for H&E staining correspond to 600 μm and 60 μm of core and enlarged tumor images, respectively. For IHC staining of DUSP6, scale bars correspond to 100 and 60 μm of `core and enlarged tumor images, respectively. **e, f** Clonogenic assays (**e**) and immunoblotting (**f**) of KRAS G12D eIF2α$^{S/S}$ and eIF2α$^{A/A}$ mouse lung tumor cells treated with either scrambled siRNAs or a mix of four different DUSP6 siRNAs. Graphs represent quantitative analyses from three biological replicates. **a–f** Data represent mean ± SEM. Significance in differences between two datasets was determined using two-tailed unpaired $t$-test. $P$ values are indicated in the bar graphs (ns non-significant).

control-treated mice ($n = 9$) perished within 45 weeks of treatment, while only two out of six ISRIB-treated mice perished within the same period (Fig. 6f). Collectively, the findings demonstrated the anti-tumor effects of ISR inhibitors on mutant KRAS lung tumor growth in mice and their therapeutic potential for the treatment of lung cancer.

## Discussion

Our study portrays the adaptive PERK/p-eIF2α branch of ISR as an essential component of tumorigenesis and a valid target of therapeutic intervention for mutant KRAS lung cancer treatment. The clinical relevance of our findings is supported by the examination of a large cohort of 928 archival primary human LUADs, which revealed that p-eIF2α is elevated in a way that correlates with local invasiveness, with high-risk tumor subtypes, and with tumor cell proliferation (Fig. 1; Supplementary Fig. 1). Levels of p-eIF2α are highest in areas of solid pattern invasive growth, which is known to be a particularly high-risk prognostic feature[34]. The additional virulence of tumors with high levels of p-eIF2α is likely to be the result of this enhanced cellular invasion and proliferation, which are hallmark behaviors that would be expected to influence tumor growth pattern; it is, therefore, unsurprising that while higher levels of p-eIF2α predict poor outcome in univariate models they lose significance in multivariate models which include growth pattern.

Using a mouse model of lung cancer, we demonstrate that the translational repression of DUSP6 by p-eIF2α is an important mechanism of mutant KRAS tumorigenesis (Fig. 3c). DUSP6 exhibits anti-tumor effects in human LUAD cells[35] while its decreased expression is linked to poor prognosis and increased sensitivity of LUADs to MAPK inhibition[36]. Translational inhibition of DUSP6 accounts for the stimulation of p-ERK by p-eIF2α since DUSP6 downregulation by siRNAs restored p-ERK and increased the survival of KRAS G12D eIF2α$^{A/A}$ cells to equal levels of KRAS G12D eIF2α$^{S/S}$ cells (Fig. 3e, f). A recent study showed that hyperactivation of p-ERK by the pharmacological inhibition of DUSPs can exert anti-proliferative and anti-tumor effects in mutant KRAS LUAD cells[37]. However, stimulation of p-ERK by the translational suppression of DUSP6 is below the threshold required for the induction of anti-proliferative effects in the mouse KRAS G12D lung tumors (Fig. 3e, f). In LUAD patients, we observed a significant association between p-eIF2α and p-ERK at the single-cell level in several primary tumors, supporting our proposed mechanistic chain of events within individual tumors (Supplementary Fig. 4). In addition to p-ERK stimulation, we pinpoint roles for adaptive ISR in driving secondary changes in the transcriptome resulting in further

alterations to signaling and metabolic pathways with established roles in cancer. Specifically, UR analysis of the RNA-seq data indicated a role of p-eIF2α in the activation of pro-tumorigenic pathways under the control of *CTNNB1, LHX1, HIF1A,* and pro-inflammatory IL6 pathway (Fig. 2g)[38–42]. In contrast, p-eIF2α was predicted to negatively regulate tumor suppressors like the homeobox protein *HOXA10*, estrogen-related receptor alpha (*ESRRA*), polycomb group protein *ASXL1,* and mitofusin 2 (*MFN2*) function (Fig. 2g)[43–46]. Gene set enrichment analyses (GSEA) suggested the involvement of the p-eIF2α-dependent genes in the stimulation of growth factor receptor signaling, epithelial cell proliferation, and mesenchymal cell differentiation (Supplementary Fig. 12). In addition, GSEA highlighted an inverse relationship between p-eIF2α and the expression of genes involved in mitochondrial respiration (Supplementary Fig. 12). Considering that increased p-ERK favors aerobic glycolysis[47], p-eIF2α may play a role in this metabolic process through its ability to stimulate p-ERK as well as decrease mitochondrial respiration and oxidative phosphorylation in mutant KRAS tumors.

The transforming properties of mutant KRAS are accompanied by an increased exposure of cells to genotoxic, proteotoxic, and metabolic stress as a result of the disruption of normal proliferation and tissue homeostasis[48,49]. The successful growth of mutant KRAS tumors depends on the action of adaptive mechanisms to cope with these stresses and maintain proliferation[48,49]. We show that mutant KRAS upregulates the PERK/p-eIF2α branch in human LUAD cells to promote the survival and growth in response to stress (Supplementary Figs. 6, 7). This is consistent with previous work showing that adaptive ISR plays an important role in the tolerance of mutant KRAS-transformed embryonic fibroblasts to hypoxic stress[50] and adaptation of human LUAD cells to nutritional stress[51].

Mutant KRAS causes a persistent activation of the PERK/p-eIF2α pathway (Supplementary Fig. 6), which accounts for the higher susceptibility of mutant than WT KRAS lung cancer cells to ISR inhibition (Fig. 6a, b; Supplementary Fig. 7). ISRIB counteracts ISR activation by pre-disposing its target eIF2B into an active state that becomes resistant to inhibition by p-eIF2α[52,53]. Low p-eIF2α results in ISR that is weakly antagonized by ISRIB[52,53], thus explaining the resistance of WT KRAS tumor cells to ISRIB treatments in mice (Fig. 6a). Contrary to ISRIB, the PERK inhibitor GSK2606414 reduced the growth of WT KRAS tumors in mice, although this effect was not statistically significant ($p = 0.116$) (Fig. 6a). The anti-tumor trend of the PERK inhibitor in WT KRAS tumor cells with low ISR may be explained by the implication of PERK in p-eIF2 independent pathways like the stimulation of the nuclear factor erythroid

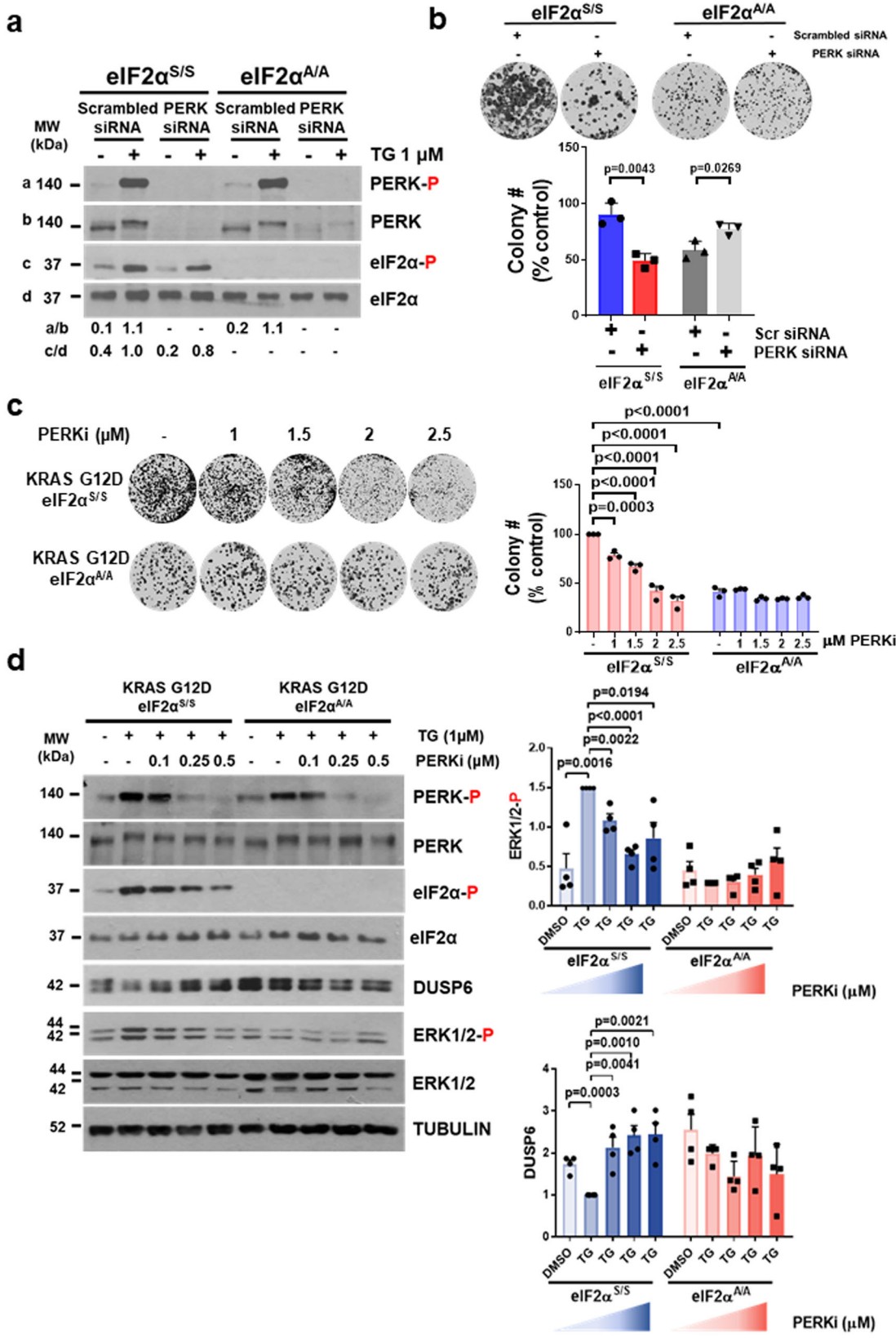

2-related factor 2 (NRF2)-HIF pathway in pancreatic and lung cancers[54].

ISR's ability to integrate multiple tumor regulatory pathways highlights the potential therapeutic value of its pharmacological inhibition for the treatment of mutant KRAS lung cancer. ISRIB and an emerging generation of ISRIB derivatives have shown remarkable results in the treatment of cognitive disorders[4]. A salient property of ISRIB is the ability to specifically blunt ISR and inhibit tumor formation in mice without toxic side-effects even after prolonged treatments[55,56]. The lack of tissue toxicity may provide ISRIB with a considerable advantage over conventional therapies like treatments with MEK inhibitors, which are

**Fig. 4 PERK dictates the pro-survival and translational effects of p-eIF2α in KRAS G12D tumor cells. a** Immunoblotting of mouse KRAS G12D eIF2α$^{S/S}$ and eIF2α$^{A/A}$ lung tumor cells subjected to treatments with either scrambled siRNAs or a mix of four different PERK siRNAs followed by 1 μM thapsigargin (TG) treatment for 1.5 h, ($n = 1$). **b** Clonogenic assays of KRAS G12D eIF2α$^{S/S}$ and eIF2α$^{A/A}$ cells subjected to either scrambled siRNAs or PERK siRNAs treatments. The graph represents data from three biological replicates. **c** Colony-forming efficacy of mouse KRAS G12D eIF2α$^{S/S}$ and eIF2α$^{A/A}$ cells after treatment with the indicated concentrations of the PERK inhibitor (PERKi) GSK2606414. Graphs represent data from three biological replicates. Data represent mean ± SEM. Significance in differences between datasets was determined using two-sided one-way Anova, Tukey's multiple comparison test. P values are indicated on the bar graph. **d** KRAS G12D eIF2α$^{S/S}$ and eIF2α$^{A/A}$ cells were untreated or pre-treated with 1 μM thapsigargin (TG) for 30 min followed by treatments with increasing concentrations of PERKi for 1 h. Protein extracts (50 μg) were subjected to immunoblotting for the indicated proteins. Quantifications of blots were performed from four biological replicates. Phosphorylated ERK (ERK-P), PERK (PERK-P), and phospho-eIF2 (eIF2α-P) were normalized to corresponding total protein whereas DUSP6 expression was normalized to ACTIN or TUBULIN. **b, d** Data represent mean ± SEM. Significance in differences between two datasets was determined using two-tailed unpaired t-test. P values are indicated on the bar graph.

associated with many adverse side effects in the clinics[57]. Because mutant KRAS-driven lung cancers are largely resistant to therapy, ISR inhibitors like ISRIB are promising agents for the treatment of this common deadly mutant KRAS-driven malignancy and may be applicable to other common cancers with KRAS mutations. The recent development of highly specific irreversible inhibitors for KRAS G12C holds promise for the treatment of LUAD and other cancer forms with this type of KRAS mutation[58]. Nevertheless, the larger fraction of LUAD contains KRAS mutations different from G12C, such as G12D, which is the most common KRAS mutation (56%) among non-smokers[59]. Pharmacological inhibition of ISR potently impairs lung tumor growth with KRAS G12C as well as G12D, supporting a broader anti-tumor effect on cancers with different KRAS mutations. In summary, therefore, by shedding more light on the role of the ISR in mutant KRAS signaling and lung tumorigenesis, our work strongly suggests that innovative therapeutic approaches using ISR inhibitors may be valuable for the treatment of one of the deadliest forms of mutant KRAS-driven cancer.

## Methods

**Cell lines and treatments**. H358,H23, H1299, H1703, and LLC were maintained in RPMI 1640 medium (Wisent) supplemented with 10% fetal bovine serum (FBS, Wisent) and 1% antibiotics (penicillin/streptomycin, 100 units/mL; Life Technologies). H1299 cells were engineered to overexpress WT KRAS 4B, KRAS 4B G12C, KRAS 4B G12V, and KRAS 4B G12D by the transfection of PCDNA3.1 plasmids bearing the KRAS 4B cDNAs and selection in 500 ug/ml G418 (Gibco)[60–62]. The inserted cDNAs were verified by Sanger sequencing. The functionality of mutated KRAS cDNAs was determined by analyzing downstream signaling pathways (i.e., ERK phosphorylation) as well as by determining the interaction of KRAS proteins with BRAF using NanoBRET KRAS/BRAF Interaction Assay (Promega) or by using the KRAS Activation Assay Kit (Cell Biolabs, San Diego, CA). H1703 cells over-expressing either green fluorescence protein (GFP)-WT KRAS or GFP- KRAS G12C were established by infection with retroviruses expressing the GFP tagged KRAS 4B cDNAs[33]. Primary KRAS G12D eIF2α$^{S/S}$ and eIF2α$^{A/A}$ lung tumor cells were isolated from mice at 20 weeks of lung tumor formation. Mouse lung lobes were washed with ice-cold sterile phosphate-buffered saline (PBS), chopped into ~1 mm$^3$ pieces, and incubated with 1 mg/mL collagenase in serum-free DMEM media 2 h at 37 °C under continuous rotation. The homogenate was centrifuged at $200 \times g$ for 3 min, the pellet was washed three times with ice-cold PBS and suspended in 2 mL of trypsin-EDTA solution (Life Technologies) for 5 min at 37 °C under rotation. After three washes in ice-cold PBS and centrifugation at 200xg for 3 min, the pellet (~1 × 10$^7$ cells) was suspended in RPMI 1640, 10% FBS, antibiotics (100 units penicillin/streptomycin), 0.075% Sodium Bicarbonate NaHCO$_3$ (Life Technologies), 1X essential amino acids (Life Technologies), 1X non-essential amino acids (Life Technologies). GFP-positive KRAS G12D eIF2α$^{S/S}$ and eIF2α$^{A/A}$ cells were sorted by flow cytometry and maintained in the same RPMI 1640 media. Downregulation of mouse DUSP6 or PERK was performed by treatments with a mix of 4 siRNAs (Dharmacon) containing the sequences listed in Supplementary Table 1. Colony formation assays were performed with 10$^3$ cells subjected to anti-tumor treatments for 14 days as indicated in figure legends. Cells were fixed in 4% v/v paraformaldehyde and stained with 0.2% w/v crystal violet. Colonies were scored using an automated cell colony counter (Gel-Count; Oxford Optronix). (E)-2-benzylidene-3-(cyclohexylamino)-2,3-dihydro-1H-inden-1-one (BCI) inhibitor was obtained from Millipore Sigma, GSK2606414 from MedKoo, ISRIB from Selleck Chemicals.

**Flow cytometry analysis**. Cells were plated the day before at ~20% confluency to achieve 80–90% confluency in six-well plates at 72 h of treatment. PERK inhibitor

was added the next day and media was refreshed with the inhibitor every 24 h. After treatment, the cells were lifted by incubating with phosphate buffer saline (PBS) plus 0.5 mM EDTA for 5 min at 37 °C and an equal volume of media with 10% FBS. Cells were centrifuged at $500 \times g$ for 5 min and washed two times with ice-cold PBS. Cells were resuspended with ice-cold 70% ethanol in PBS and stored at −20 °C for at least 30 min. For propidium iodide (PI) staining, cells were spun down at $1000 \times g$ for 5 min and washed twice with ice-cold PBS. Cells were resuspended in PI buffer (4 μL Triton X-100, 40 μg PI, 0.5 mg RNAse A, up to 1 mL with PBS), incubated at 37 °C for 30 min followed by FACS analysis using BD LSRFortessa flow cytometer. FACS data were collected using FACSDiva software and analyzed using FlowJo software.

**Urethane and KRAS G12D induced lung tumorigenesis**. Four-week-old male and female C57BL/6 mice, which were either proficient (eIF2α$^{S/S}$) or haplo-insufficient for p-eIF2 (eIF2α$^{S/A}$;[26]) were subjected to a single intraperitoneal injection of urethane (Sigma) at 1 g/kg in pups between 21 and 28 days old[63]. Induction of lung tumorigenesis in KRAS$^{+/LSL-G12D}$;fTg/0;eIF2α$^{S/S}$ and KRAS$^{+/LSL-G12D}$;fTg/0;eIF2α$^{A/A}$ C57BL/6 mice was performed by the intratracheal intubation of CRE- and TP53 shRNA-expressing lentiviruses[18,64] and lung tumor formation was monitored by Ultrasound Imagining using the VisualSonics VEVO 3100 high-frequency ultrasound apparatus according to manufacturer's specifications[64].

**Xenograft tumor assays**. Cells were suspended in 50% Matrigel Matrix GFR (Corning) in PBS prior to injection. Cells (1 × 10$^6$ in 0.1 ml) were injected sub-cutaneously in the flanks of 8-week-old female BALB/c nude mice (Charles River Laboratories). Tumor growth in NOG and SCID mice was measured with digital calipers two times per week, and the volume calculated by the formula: tumor volume [mm$^3$] = [(length [mm]) × (width [mm])$^2$]/2. Mouse treatment with PERK inhibitor GSK2606414 or ISRIB by oral gavage consisting of 0.5% HPMC solution and 0.1% Tween 80, pH = 4.0 at concentrations indicated in the figures[55,65]. Orthotopic transplantation of LLC cells (2 × 10$^5$) in C57BL/6 mice was performed by intratracheal intubation[66].

**Guidelines of ethical conduct in mouse work**. The animal studies were performed in accordance with the Institutional Animal Care and Use Committee (IACUC) of McGill University and procedures were approved by the Animal Welfare Committee of McGill University (protocol #5754).

**Polysome profiling, RNA isolation, and real-time qPCR**. For polysome profiling analysis cells were lysed in 10 mM Tris HCl, pH 7.4, 150 mM NaCl, 0.5 % NP40, 10 mM MgCl$_2$, 100 μg/ml cycloheximide, 2 mM dithiothreitol, 100 U/ml RNA Guard and fractionated on 10–55% sucrose gradients by ultracentrifugation (SW41 rotor; Beckman 30,000 rpm, 3 h at 4 °C)[67]. The gradients were prepared with the ISCO model 160 Gradient Former and fractionated into 500 μl fractions using the ISCO density gradient fractionation system Foxy Jr. Fraction Collector while measuring the absorbance at 254 nm. Total RNA and polyribosomal RNA (1 μg) isolated by Trizol (Thermo Fisher Scientific) was subjected to reverse transcription (RT) with 100 μM oligo (dT) primer using the SuperScript III Reverse Transcriptase kit (Invitrogen) according to the manufacturer's instruction. Real-time (quantitative) PCR was performed using the SensiFast SYBR Lo-ROX kit (Bioline) with primers listed in Supplementary Table 1. The qPCR assays included primers for mouse GAPDH and actin mRNAs as internal controls according to the Minimum Information for Publication of Quantitative Real-Time PCR Experiments (MIQE) guidelines[68].

**RNA-seq data analysis**. Total RNA of KRAS G12D eIF2α$^{S/S}$ and eIF2α$^{A/A}$ cells (four replicates each) was isolated with Trizol (Thermo Fisher Scientific) and RNA-Seq libraries were prepared following the TruSeq Stranded Total RNA protocol (Illumina) according to the manufacturer's instructions and 50 base single-end reads were obtained using a HiSeq2500 system in Rapid Mode (Illumina). The resulting reads were mapped to the mm10 genome assembly using HISAT and quantified using

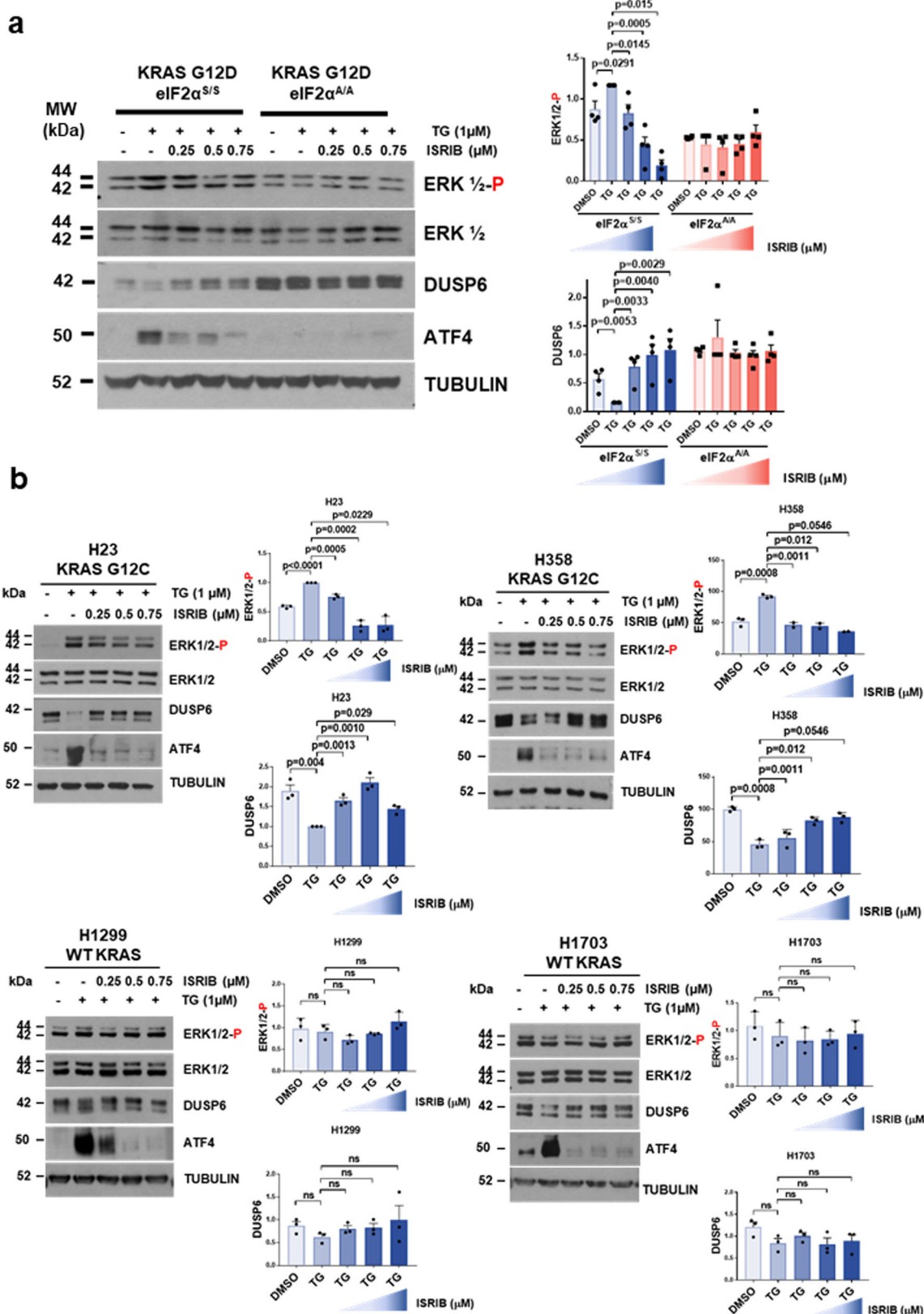

default settings[69,70]. Differential expression was performed using the random variance model as implemented in the anota2seq package (1.8.0)[71]. Genes with absolute log (FC) > 1 and False Discovery rates (FDRs) < 0.05 were considered differentially expressed. Upstream Regulator analysis was performed as implemented in IPA. Briefly, this analysis is based on the prior knowledge of expected effects between transcriptional regulators and their target genes (Ingenuity® Knowledge Base). The analysis examines how many known targets of each transcription regulator are present in the dataset and compares their direction of change to what is expected from the literature to predict likely transcriptional regulators. If the observed direction of change is mostly consistent with an activation state of the transcriptional regulator, a prediction is made about that activation state. Gene set enrichment analysis (GSEA v4.0.3, Broad Institute) was performed on all genes ranked according to fold change, using the Gene Ontology geneset v5.2 (MSigDB)[72]. The number of permutations was 1000 and only sets containing between 15 and 500 genes were retained.

**Fig. 5 ISR inhibition antagonizes p-eIF2α function in mutant KRAS lung tumor cells. a**, **b** Mouse KRAS G12D eIF2α$^{S/S}$ and eIF2α$^{A/A}$ cells (**a**) or human LUAD cell lines with either WT KRAS or KRAS G12C (**b**) were left untreated or pre-treated with 1 μM thapsigargin (TG) for 30 min followed by treatments with increasing ISRIB concentrations for 1 h. Protein extracts (50 μg) were subjected to immunoblotting for the indicated proteins. Quantifications of blots for phosphorylated ERK (ERK-P) and DUSP6 in eIF2α$^{S/S}$ and eIF2α$^{A/A}$ cells were obtained from four biological replicates. ERK-P was normalized to total ERK whereas DUSP6 expression to Tubulin. One biological replicate consisted of eIF2α$^{S/S}$ and eIF2α$^{A/A}$ samples analyzed on the same blot shown in a whereas the rest of the biological replicates consisted of eIF2α$^{S/S}$ and eIF2α$^{A/A}$ samples derived from the same sets of experiments analyzed on separate blots but processed in parallel. ATF4 was used as a marker of the antagonistic effects of ISRIB on p-eIF2-mediated mRNA translation in the tumor cells. Data represent mean ± SEM. Significance in differences between two datasets was determined using two-tailed unpaired *t*-test. *P* values are indicated on the bar graph.

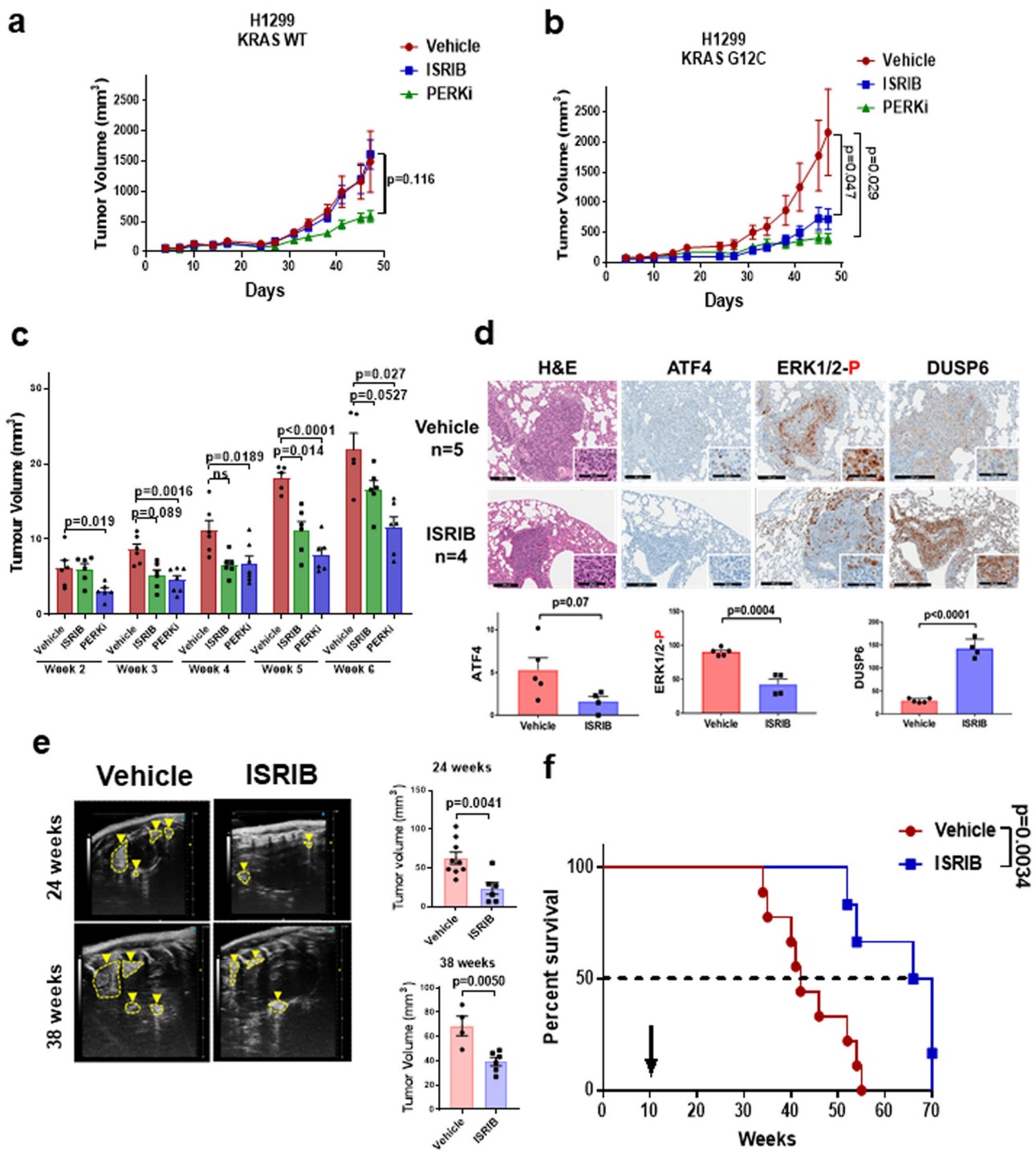

**Fig. 6 Pharmacological inhibition of ISR impairs KRAS lung tumor growth. a, b** H1299 cells overexpressing either wild type KRAS (**a**) or KRAS G12C (**b**) were transplanted subcutaneously in nu/nu mice followed by treatments with vehicle control ($n = 8$ for wild type KRAS; $n = 6$ for KRAS G12C), 10 mg/kg ISRIB ($n = 10$ for wild type KRAS; $n = 8$ for KRAS G12C) or 150 mg/kg PERK inhibitor GSK2606414 (PERKi) ($n = 10$ for wild type KRAS; $n = 6$ KRAS G12C). Data represent mean ± SEM, two-sided one-way ANOVA, Dunnett's multiple comparison test, $P$ values are indicated on the graph. **c, d** LLC cells were orthotopically injected into the lungs of immune-competent C57BL/6 mice. Mice were subjected to treatments on day 12 after the intratracheal intubation of LLC cells. The graph in panel c indicates lung tumor volume at the indicated weeks of tumor growth as analyzed by ultrasound imaging of mice treated with vehicle ($n = 6$), 10 mg/kg ISRIB ($n = 6$), and 150 mg/kg PERKi ($n = 7$). Data represent mean ± SEM, two-sided one-way ANOVA, Dunnett's multiple comparison test, $P$ values are indicated on the graph. **d** H&E staining along with the expression of nuclear ATF4, nuclear p-ERK and cytoplasmic DUSP6 in vehicle-control or ISRIB-treated LLC tumors at the sixth week of treatment. Scale bars correspond to 200 and 60 μm of core and enlarged tumor images, respectively. **e, f** Immune-competent mice bearing KRAS G12D lung tumors were subjected to treatments with either vehicle ($n = 9$) or 10 mg/kg ISRIB ($n = 6$). Ultrasound imaging in panel e indicates lung tumor formation in live mice at 24 or 38 weeks after treatment initiation. Graphs indicate tumor volume assessed by ultrasound imaging. The survival curve in (**f**) refers to immune-competent mice bearing KRAS G12D tumors treated with either vehicle control ($n = 9$) or 10 mg/kg ISRIB ($n = 6$). Arrowhead indicates initiation of drug treatment 10 weeks after KRAS G12D induction in the lungs by the intratracheal intubation of CRE-expressing lentiviruses, at which point detectable lung tumors were formed (Supplementary Fig. 11b). **d, f** Data represent mean ± SEM. Significance in differences between two datasets was determined using two-tailed unpaired $t$-test. $P$ values are indicated on the bar graph.

**Protein extraction and immunoblotting**. Cells were washed twice with ice-cold PBS and proteins were extracted in ice-cold lysis buffer containing 10 mM Tris-HCl, pH 7.5, 50 mM KCl, 2 mM $MgCl_2$, 1% Triton X-100, 3 μg/ml aprotinin, 1 μg/ml pepstatin, 1 μg/ml leupeptin, 1 mM dithiothreitol, 0.1 mM $Na_3VO_4$, and 1 mM phenylmethylsulfonyl fluoride. Extracts were kept on ice for 15 min, centrifuged at $10,000 \times g$ for 15 min (4 °C), and supernatants were stored at −80 °C. Proteins were quantified by Bradford assay (Bio-Rad). The expression of different proteins was tested in parallel by loading 50 μg of protein extracts from the same set of samples on two identical sodium dodecyl sulfate (SDS)-polyacrylamide gels. After protein transfer to Immobilon-P membrane (Millipore), the two identical blots were cut into smaller pieces based on the size of proteins to be tested. One piece was probed for the phosphorylated protein of interest whereas the other identical piece for the corresponding total protein. The antibodies used for immunoblotting are listed in Supplementary Table 2. Proteins were visualized by enhanced chemiluminescence (ECL) according to the manufacturer's specification (Amersham Biosciences). Quantification of bands in the linear range of exposure was performed by the ImageJ 1.51e software (NIH, Maryland, USA).

**Preparation of TMAs**. TMAs were constructed from a continuous series of archival primary resected LUADs obtained by University Hospital Leicester NHS Trust between 1998 and 2015. Samples were excluded if the patient had any previous lung cancer diagnosis. Whole diagnostic H&E sections were reviewed, and 3× representative tumor cores (1 mm) were taken in triplicate from FFPE blocks and embedded in a total of 23 acceptor blocks. Outcome and pathological data of patients were collected from local and national databases. TMAs were sectioned at 4.5 μM. All TMAs were H&E stained, and patterns ascribed to individual cores according to WHO guidelines; where necessary, whole sections images from donor blocks were examined to confirm growth pattern. This study was approved by the Northampton Research Ethics Committee (reference 14/EM/1159) and University Hospitals Leicester NHS Trust Research and Innovation Department (reference UHL 11363).

**Histology and immunohistochemistry (IHC)**. Mouse tissues were fixed in 10% buffered formalin phosphate, paraffin embedded, and sectioned. Paraffin was removed from the sections after treatment with xylene, rehydrated in graded alcohol, and used for H&E staining and immunostaining. Antigen retrieval was performed in sodium citrate buffer. Primary antibodies were incubated at 4 °C overnight and secondary antibodies were incubated at room temperature for 90 min (antibodies are listed in Supplementary Table 2). Sections were counterstained with 20% Harris modified hematoxylin (Thermo Fisher Scientific), mounted in Permount solution (Thermo Fisher Scientific), and scanned using an Aperio Scanscope AT Turbo scanner (Leica biosystems). Quantification of stained sections was performed using Aperio Imagescope software (Leica Biosystems) according to the manufacturer's specifications.

For human specimens, IHC was used to examine p-eIF2 and cytokeratin expression in a duplex chromogenic assay and to examine p-eIF2, p-ERK, and cytokeratin expression in a multiplex fluorescent assay. IHC was performed on the Roche DISCOVERY Ventana® platform using Roche DISCOVERY reagents. Sections were de-paraffinized and antigen retrieval (64 min, 95 °C, pH 9.0) was performed. Endogenous peroxidase was inhibited, and non-specific Ig binding was blocked using Goat Ig, for 20 min each.

For the duplex assay, p-eIF2α primary antibody was incubated on slides and detected using a secondary antibody. The Roche AMP HQ kit was used to amplify p-eIF2 DAB staining. Antibody denaturation (8 min, 100 °C, pH 6.0) and neutralization steps (20 min, DISC inhibitor) was performed prior to further

blocking (12 min, Goat Ig) and cytokeratin antibody incubation. A purple detection kit was used to detect cytokeratin AE1/AE3. The slides were counterstained with haematoxylin and sections were dehydrated and mounted. Primary and secondary antibodies are listed in Supplementary Table 3.

Duplex stained slides were scanned at x40 on the Hamamatsu NanoZoomer-XR C12000. Slide images were imported, de-arrayed, and analyzed using the Visiopharm® digital pathology platform. An app was developed to detect and outline tumor areas using the purple cytokeratin stain. A further app identified individual cells and generated a H-score based on DAB/purple intensity within the tumor area (H-score = $3 \times \%$ strong staining $+ 2 \times \%$ moderate staining $+ \%$ weak staining). Quantitative H-scores were generated from digital TMA images using Visiopharm® software, based on the intensity and proportion of cytoplasmic p-eIF2 staining within tumor cells as identified by cytokeratin staining. Visiopharm® H-scores were validated against a manually scored TMA. For p-eIF2 analysis, patients were divided into two groups based on a positive/negative cut-off value determined through correlation of IHC images and H-scores; cores with an H-score of <6 were deemed to be immunohistochemically negative. This automated method was validated against manual H-scoring of a representative TMA (120 donor cores), giving a Spearman's Rho of 0.939, $p < 0.001$. For each patient, the median of p-eIF2 H-scores from up to 3 cores/tumor was used.

For the fluorescent multiplex assay, antibodies were applied in the sequential order in Supplementary Table 3 with an antibody denaturation (8 min, 100 °C, pH 6.0) and neutralization (20 min. DISC inhibitor) steps in between. The slides were scanned at ×20 in the Akoya Vectra® and tumor regions were analyzed using Akoya inform Advanced Image Analysis software. Tissue and cells were segmented based on the fluorescent channels and mean pixel intensity data were collected for each marker at the single-cell level.

**Statistical analysis of patients' data**. For patient data, statistical analysis was performed using RStudio (1.0.153). Spearman's rank correlation was used to validate Visiopharm® H-scores and assess the relationship between p-eIF2 and Ki67. p-eIF2 was directly measured and quantified by IHC in 928 human LUADs, providing a broad range of staining intensities. H-scores were generated using Visiopharm® software based on the intensity and proportion of cytoplasmic p-eIF2 staining. The automated scores were validated against manual H-scoring of a representative TMA (120 donor cores), giving a Spearman's Rho of 0.939, $p < 0.001$. Patient survival was visualized by Kaplan–Meier plots and significance assessed by a log-rank test and Cox Proportional regression for univariate survival models. The associations between patient survival and p-eIF2 were examined using overall, cancer-specific, and recurrence-free survival endpoints. For each patient, the median of the 3-core p-eIF2 H-scores was used. Associations between p-eIF2 and histological pattern/WHO type were assessed using non-parametric Mann–Whitney–Wilcoxon and the Kruskal–Wallis tests.

**Reporting summary**. Further information on research design is available in the Nature Research Reporting Summary linked to this article.

## Data availability
The RNA seq data generated in this study have been deposited in the Gene Expression Omnibus (GEO) GSE155238. The remaining data are available within the Article and its Supplementary Information and source data. Source data are provided with this paper.

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

## Acknowledgements

We thank R. Kaufman (Sanford Burnham Medical Research Institute, La Jolla) for fTg/0; eIF2α$^{S/S}$ and fTg/0;eIF2α$^{A/A}$ mice; Miss T. Yang for technical assistance in mouse breeding; S. Huang (McGill University) for H1703, H1299, and H23 cells; M. Witcher (McGill University) for H358 cells; S. Wing (McGill University) for LLC cells; A. Willis and M. Bushell for helpful comments on the manuscript; the Research Pathology Facility at the Lady Davis Institute-Jewish General Hospital for IHC imaging; Expert histological assistance in human tissue experiments was provided by Catherine Ficken, Vendula Jones, and Madhumita Das, and Dr Juvenal Baena assisted in tumor morphological assessment. The work was supported by the Canadian Institutes of Health Research (CIHR) to A.E.K. (MOP-137113, MOP-38160, PJT-168864) and I.T. (MOP-363027) National Institutes of Health (NIH) to M.H. (DK060569, DK053307) as well as the Swedish Research Council to O.L. I.T. is a recipient of Le Fonds de recherche du Québec (FRQS)—Santé Junior 2 award and O.L. is a Wallenberg Academy Fellow. N.G. is a recipient of Faculty of Medicine/ McGill University and FRQS studentship awards.

## Author contributions

N.G. collected, analyzed, and interpreted data in Figs. 2c, 3, 4, 5, 6 Supplementary Figs. 5, 6, 7, 8, 9, 10, 11, partially wrote and edited the manuscript; S.W. collected and analyzed data in Figs. 2b, c, e, 3b, 6; B.W., L.O., and A.T. collected and analyzed data in Fig. 1, Table 1, and Supplementary Figs.1, 4; J.K. collected data in Figs. 3b, 4d, 5 Supplementary Figs. 6, 7, 9; C.D. collected and analyzed data in Figs. 2d, 3a, d, Supplementary Fig. 2; U.K. and N.A. collected, analyzed data in Supplementary Fig. 3, and assisted in mouse breeding; M.J. collected data in Figs. 3e, 4d, 5a, Supplementary Figs. 6a, c, 7b; H.P. analyzed and interpreted data in Fig. 2d and Supplementary Fig. 2; M.B. provided reagents; K.M., M.H., and I.T. assisted in data analyses and interpretation; V.H. and O.L. analyzed and interpreted data in Fig. 2f, g, Supplementary Fig. 12; J.L.Q. designed the study for the prognostic value of p-eIF2α, interpreted data in Fig. 1, Table 1, Supplementary Figs. 1, 4, partially wrote and edited the manuscript; A.E.K. designed the study, analyzed and interpreted the data, wrote and edited the manuscript.

## Competing interests

The authors declare no competing interests.
