## [Peer Review File · Nature Communications]

REVIEWER COMMENTS

Reviewer #1 (Remarks to the Author):

In this study Ghaddar et al address a role for the Integrated Stress Response (ISR) in KRAS driven lung adenocarcinoma (LUAD). The impact of ISR activation in cancer is a rapidly developing area. Recent studies have linked the ISR to the progression of several cancers including breast cancer (Jewer M et al, 2020), prostate cancer (Nguyen HG et al, 2018) and LUAD (Gwinn DM et al, 2018, Albert A et al, 2019). In this particular study the authors demonstrate, using elegant in vivo models, a requirement for ISR activation to enable efficient KRAS-driven lung tumor growth. Using patient samples and data sets they demonstrate an association between p-eIF2a (ISR activation) and patient outcome. In terms of mechanism, the authors demonstrate a linkage between ISR activation (as defined by p-eIF2a) and phosphorylation of ERK. They conclude that ISR activation permits increased phosphorylation of ERK by decreasing DUSP6 expression in KRAS lung adenocarcinoma cells. KRAS lung cancer cells expressing eIF2sS51A variant which cannot trigger an ISR demonstrated higher DUSP6 expression and lower p-ERK. The authors also conclude that PERK, an endoplasmic reticulum anchored transmembrane receptor, is responsible for mediating phosphorylation of eIF2a and demonstrate that treatment with an inhibitor reported to target PERK (GSK2606414) blocks eIF2a phosphorylation, reduces p-ERK and increases DUSP6. In vivo a therapeutic benefit in terms of reducing tumor growth and increasing survival is demonstrated using both ISR inhibitor ISRIB. Based on these findings the authors suggest that ISRIB could represent a good treatment strategy for LUAD.

General Comments

The differences obtained in tumor growth, frequency and size between KRASG12D eIF2a S/S and KRASG12D eIF2a A/A are clear and obvious. How ISR activation is achieving this is less clear. Although the authors propose a mechanism ISR-DUSP6-p-ERK the linkage between the functional effects mediated by ISR activation and the proposed mechanism is a little under developed. In this study the authors present PERK solely as a mediator of the ISR but PERK activation is indicative of the UPR. While it may be hard to define where the UPR stops and the ISR begins it seems remiss to not acknowledge that PERK is a key mediator of the UPR. Indeed several experiments addressing mechanism utilize the ER stress inducer Tg. The in vivo models tracking tumor growth appear to block ISR activation from the very beginning. As the authors state in the introduction, a particular problem with LUAD is late detection. What happens if you model this scenario, rather than treat from the start, allow the tumor to grow to a particular size and then treat with ISRIB, do you observe still tumor regression? I do see that the drug treatment in the survival curve (Fig 6C) was not initiated until 10 weeks post tumor initiation but all studies assessing tumor volume (6a & b) were treated with PERKi or ISRIB from the start? The in vivo data demonstrating the efficacy for ISRIB treatment and PERK inhibitor presented in figure 6 is striking but is not accompanied by data demonstrating on-target efficacy. Can the authors demonstrate on-target effects, is p-eIF2a reduced in these tumors? In this scenario it would also be helpful to assess if there are also differences in p-ERK and DUSP6 expression. GSK2606414 is described as "a potent and selective PERK inhibitor". This 2017 study (<https://www.ncbi.nlm.nih.gov/pmc/articles/PMC5442476/>) found GSK2606414 in addition to reducing PERK activity was a potent RIPK1 inhibitor suggesting this inhibitor is not selective to PERK. Based on this findings with GSK2606414 should be replicated using additional selective approaches such as genetic manipulation of PERK expression. A recent study by Unni AM et al (eLIFE 2018) suggests that DUSP6 upregulation is required in LUAD to ensure ERK activity stays within a certain threshold. They demonstrate a reduction in DUSP6 levels associates with an increase in p-ERK and cell death. In this study the authors conclusions are slightly different they suggest a decrease in DUSP6 expression mediated via the ISR is required for efficient tumor growth. It would be good to place the findings of this study in context with the previously published literature in the discussion.

Specific comments

Figure 2

Is the RNA seq data set available? I didn't notice a statement pertaining to data availability in the manuscript.

Figure 3

3b. A blot demonstrating a functional inhibition of the ISR e.g. ATF4 would add to this figure panel. Also an explanation of why serum deprivation conditions were used would be useful in the text. Is it to activate the UPR hence increase PERK signaling?

Figure 4

4a. Blots verifying the functional effect of GSK2606414 should be included to demonstrate a reduction in PERK phosphorylation and a reduction in eIF1a-p in eIF2aS/S. Given the issues surrounding the selectivity of GSK2606414 an additional approach e.g. knockdown of PERK via siRNA or shRNA and assessment of the outcome on DUSP6, p-eIF2a and p-ERK should be included. Is the decrease in colony formation a result of increased cell death?

4b. There is basal activation of PERK in the eIF2a S/S cells as demonstrated by the detection of p-PERK and p-eIF2a. What is the outcome if eIF2a S/S cells are treated with PERKi without the addition of Tg? Do you see an increase in DUSP6 levels and a decrease in p-eIF2a?

Figure 5

5a. Similar to my comment above what is the outcome on eIF2a, DUSP6 and ERK in eIF2a S/S cells if you add ISRIB alone and not in combination with Tg? I understand that addition of Tg is a way to "boost" PERK activation but it may not accurately model events in vivo. The level of PERK activation instigated by Tg could be many fold higher than that achieved in vivo, in addition Tg treatment may activate additional cellular pathways not triggered in vivo. It is also interesting to note that the eIF2a A/A cells have endogenously high levels of ATF4 do the authors have any thoughts on why this is the case? If an ISR is not functional in this setting it seems surprising to see high ATF4.

Figure 6

Connecting the proposed mechanism to the functional in vivo observations would really strengthen this study. While the effect of ISRIB on tumor growth is clear whether this is achieved via the proposed pathway is less developed. Evidence of on-target effect of the inhibitors in Fig 6 should be provided i.e. p-eIF2a blots and evidence of ISR activation (ATF4), DUSP6 expression changes and p-ERK expression in tumors from this in vivo experiment would be a good starting point. If BCI is used in vivo do you observe a reduction in tumor growth similar to that seen with ISRIB or PERK inhibitor? Do knockdown DUSP6 LUAD cells form tumors more rapidly? In relation to the efficacy of GSK2606414 in Fig 6 H1299 WTKRAS do the authors have any thoughts why GSK2606414 would reduce tumor growth in this setting and ISRIB would not? As stated in the general comments it would be interesting to model a scenario similar to that observed in LUAD patients where tumors are not detected until a more advanced stage. What is the outcome on tumor growth if tumors are allowed to firstly establish to various sizes and then treated with PERKi or ISRIB?

Reviewer #2 (Remarks to the Author):

It has been shown in previous studies that the ISR pathway is up-regulated in many cancers. Activation of the GCN2 arm of the ISR pathway has been shown to be a mechanism through which tumor cells cope with nutrient stress but the role of the ISR pathway in nutrient replete condition has not been explored to the same degree. In this manuscript, the authors present genetic and pharmacological evidence to support a pro-up-regulated role of the integrated stress response (ISR) in KRAS mutant lung adenocarcinoma. Pharmacological inhibition of ISR suppresses tumor growth in independent models of lung cancer and increased overall survival in mice. Mechanistically, the authors identified DUSP6 to be a selective translational target of eIF2a. Activation of ISR results in translational suppression of DUSP6 and subsequently increase in ERK phosphorylation to support tumorigenesis. This effect is apparent in KRAS mutant, but not wildtype cells. This study is generally well performed. Addressing the following points will further strengthen the conclusions of this manuscript:

1. The role of DUSP6 as a downstream effector of ISR is currently correlative.

- a. Does suppression of DUSP6 expression (shRNA, sgRNA or expression of dominant negative) in eIF2aA/A cells restore cell proliferation?
- b. Does suppression of DUSP6 expression (shRNA, sgRNA or expression of dominant negative) overcome the effect of a PERK inhibitor in colony formation assays?
- c. Is DUSP6 protein expression increased in tumors from ISRIB-treated mice?
- d. Does ISRIB treatment in the absence of thapsigargin have any impact on DUSP6 expression in eIF2aS/S cells?

2. The conclusion that ISR inhibition is effective in Kras mutant and not Kras wildtype cells is not sufficiently supported by experiments performed in a single KRAS WT cancer cell line H1299. Although H1299 is wildtype for Kras, it has a missense mutation in Nras. It is surprising that a presumably general stress response pathway in cancer cells is specific to Kras mutant cells but not that of Nras. Investigating this in additional KRAS WT cell lines such as HBEC3, and additional Nras mutant cell lines such as H2087 would further support the conclusion.

3. In vivo study:

- a. Is the effect of ISRIB in vivo cytotoxic or cytostatic?
- b. Please include a marker to validate ISRIB effect in vivo, eg ATF4 IHC in vehicle vs ISRIB treated tumors

Other concerns:

1. If possible please show better pERK IHC image for Figure 3a. The signal looks very weak in general. An image like the one presented in Figure S3c shows pERK expression much more clearly.
2. DUSP6 IHC in Figure 3e looks nuclear, can the authors comment on this since DUSP6 is a cytoplasmic protein?
3. Figure 4b and c should be run on the same gel to provide direct comparison of the effect of PERKi on S/S cells to A/A cells at baseline
4. "Normal" KRAS cells was used throughout the manuscript. "Wildtype" should be used instead.
5. Supplementary figure 4: a positive relationship is not evident at all based on the data presented. It is not clear if this adds much value to the manuscript.

Reviewer #3 (Remarks to the Author):

In this manuscript Ghaddar and colleagues have taken advantage of a tumour microarray (TMA) containing biopsies from almost 1000 primary lung adenocarcinomas (LUAD). In this material they performed immunohistochemistry to evaluate the phosphorylation of the translation initiation factor eIF2 (p-eIF2 α), a key mediator of the integrated stress response (ISR). This analysis demonstrated a significant correlation between poor outcome and high p-eIF2 α levels. Likewise, high p-eIF2 α tumours correlate with aggressive histotypes such as solid tumours and more invasive disease. These observations led the authors to propose that p-eIF2 α could be used as a prognostic marker in human LUAD.

The authors went on to use both genetically and chemically-induced LUAD mouse models to further substantiate the role of eIF2 α specifically in KRAS-mutant tumours. In this context, oncogenic KRas12D activation in vivo induces less aggressive tumours in mice that carry an S51A eIF2 α transgene that prevents its phosphorylation and activation when compared to the animals overexpressing wild type eIF2 α . Among a number of transcriptionally deregulated (both up- and down-regulated) pathways the authors have focused on the MAPK signalling pathway given its central role in this disease. They conclude that in the context of an impaired ISR KRas-mutant tumours display reduced MAPK activation as a result of elevated DUSP6 levels. This is a protein phosphatase involved in MAPK negative feedback control and the authors demonstrate that it is inhibited at the translational level by activated eIF2 α .

Next, the authors utilize in vitro assays to evaluate the effect of ISR inhibitors (both the GSK2606414, an inhibitor of the eIF2 α kinase PERK, as well as ISRIB a small molecule antagonist of the translational effects of p-eIF2 α). These experiments suggest that the putative therapeutic effect of these compounds would be KRAS-mutant specific. Finally, the authors take advantage of the inducible KRasG12D mouse models to validate the therapeutic efficacy of ISRIB treatment on the growth of endogenous tumours.

I find that the biochemical characterization of the relationship between ISR activation and MAPK

control is properly executed. Yet, I believe that the implication for human disease requires further experimental support. Furthermore, The putative specificity of ISR benefit exclusively in the context of mutant KRAS needs to be better characterized.

Major comments:

1. The analysis performed on the LUAD TMA shows a clear correlation of high p-eIF2 α levels with the more aggressive histological subtypes, as well as with tumours with high invasive and proliferative markers. Yet, given the results presented later on in the manuscript in my opinion it is essential to analyse the survival differences in high/low p-eIF2 α patients harbouring different driver oncogenes. This would be particularly important in KRAS mutant patients given the specificity of the ISR inhibition and therapeutic efficacy in this subtype suggested in the manuscript.

2. As a follow up to point #1, conceptually, I have nothing against having selected KRAS-mutant LUAD to evaluate the therapeutic efficacy of ISR inhibitors. KRAS mutations define an aggressive disease with few therapeutic options. Yet I do not understand the mechanistic basis for its KRAS specificity as suggested within the manuscript. The authors claim several times throughout the text (as an example see line 190, data support the specificity of ISRIB in targeting p-eIF2 α function in mutant KRAS cells). Data presented in the manuscript include Gene set enrichment analyses suggesting the involvement of the p-eIF2 α -dependent genes in the stimulation of growth factor receptor signaling (Supplementary Fig.7). I assume, this would also apply to EGFR-mutant tumours.

Even more confounding are the experiments using the H1299 cells, both in vitro (Figure 5 & Supp Figure 5) and upon implantation in nude mice (Figure 6). H1299 are NRAS-mutant cells (Gln61Lys) and should, presumably, function through the same signalling pathways as the KRAS counterparts. This reviewer wonders if the differential response observed could be due to the overall stress present in these cell lines. Within the text the authors refer a stress phenotype, caused by increased DNA damage, proteotoxic, metabolic and oxidative stress as a consequence of oncogenic mutations. Could the authors analyse markers for some of these stress forms to evaluate a putative correlation to the response to ISR inhibitors? Do H23 and H358 cells (used in Figure 5) display higher stress markers than the non-responsive H1299? Similarly, is this also the case in sections from implanted H1299 cells overexpressing exogenous KRAS12C (Figure 6) when compared to the controls overexpressing the KRAS wild type?

3. The chemically induced carcinogenic experiment was performed on haploinsufficient eIF2 α S/A mice. Does this mean that partial eIF2 α inhibition could be therapeutically relevant in the clinic? Or does this imply some sort of dominant-negative effect of the S51A mutant?

4. The abstract indicates (line 32) "p-eIF2 α causes specific translational repression of dual specificity phosphatase 6". I am not sure what the authors mean by specific since the transcriptional variations are pleiotropic. For instance, the PTEN phosphatase is significantly down-regulated in eIF2 α -impaired cell lines as indicated in Figure 2g. Is the activity of the PI3K-Akt pathway increased? This could be assessed in the serum starvation and release (Figure 3b) as well as in sections from eIF2 α A/A tumours (Figure 3e). The PI3K pathway is also essential an essential pathway in KRas-driven LUAD as demonstrated by work from the Downward laboratory and others.

5. The potential use of compounds targeting the ISR is based on the in vivo experiments shown in Figure 6c. Yet, whether these results could be extended to patients is unclear. The drug treatment was initiated 10 weeks after tumour initiation by lentiviral infection. It will be essential to characterize what is the tumour grading at the beginning of the treatment and if there are advanced lesions at this stage. Alternatively, do the authors have data from treatments started at a later time point when the mice are present with advanced disease?

6. Likewise, to what extent this experiment with endogenous tumours (Figure 6c) mirrors the clinical scenario is unclear in another point. The rationale within the manuscript up to this point

was the comparison between high vs low eIF2 α LUAD. To this extent the authors used a transgenic strain to overexpress this factor in the context of a KRas12D oncogene and compared that to the control strain harbouring a non-phosphorylatable and therefore inactive eIF2 α S51A allele. This comparison clearly demonstrated that eIF2 α overexpression induces a more aggressive disease. Yet the experiment shown in Figure 6c is based exclusively on the inducible KRas12D strain and thereby these tumours would express "normal" eIF2 α levels. Being this the case one wonders if then even the low-eIF2 α patients would benefit from ISRIB treatment.

Have the authors ever compared the therapeutic effect of ISRIB when used in KRasG12D eIF2 α S/S vs KRasG12D eIF2 α A/A tumours? This would be important if eIF2 α expression levels are to be used as a prognostic marker as claimed by the authors.

Similarly, could the authors compare the level of eIF2 α overexpression in tumours from KRasG12D ; fTg eIF2 α S/S and the KRasG12D exclusively harbouring the endogenous eIF2 α alleles? How does this expression difference (fold change) compare to that observed in the clinic (high vs low eIF2 α patients)?

Minor points:

a) The use of the eIF2 α S51A comes with no explanation whatsoever. I interpret this allele as a non-phosphorylatable and therefore a PERK insensitive inactive control but it should be indicated for clarity.

b) A similar work reporting the use of ISR inhibitors on lung adenocarcinoma cells in vitro and upon implantation has been recently reported (DOI: 10.1158/1541-7786.MCR-19-0245).

REVIEWER #1 (Remarks to the Author):

General Comments:

Question#1. *In this study the authors present PERK solely as a mediator of the ISR but PERK activation is indicative of the UPR. While it may be hard to define where the UPR stops and the ISR begins it seems remiss to not acknowledge that PERK is a key mediator of the UPR. Indeed, several experiments addressing mechanism utilize the ER stress inducer thapsigargin (Tg).*

ANSWER: The PERK/phospho-eIF2 branch is a key mediator of UPR, which is upregulated in cells transformed by mutant KRAS (EMBO J. 2005; 24:3470-3481). In the revised version, we mention the implication of UPR in cancer (Ref. 2-4) and discuss the role of mutant KRAS in the upregulation of UPR and ISR under stress in tumors (Ref. 49,50). We used thapsigargin (TG) treatments in our experiments to obtain robust effects of PERK/phospho-eIF2 arm on DUSP6 expression and phospho-ERK in cultured lung tumor cells (see our responses below).

Question#2 *The in vivo models tracking tumor growth appear to block ISR activation from the very beginning. As the authors state in the introduction, a particular problem with LUAD is late detection. What happens if you model this scenario, rather than treat from the start, allow the tumor to grow to a particular size and then treat with ISRIB, do you observe still tumor regression? I do see that the drug treatment in the survival curve (Fig 6C) was not initiated until 10 weeks post tumor initiation but all studies assessing tumor volume (6a & b) were treated with PERKi or ISRIB from the start?*

ANSWER: We would like to clarify this important point as follows: Treatments of Lewis Lung Carcinoma (LLC) tumors in syngeneic mice (**Fig. 6c** in revised version) with ISR inhibitors were initiated 12 days after the orthotopic transplantation of the cells at which time large tumors were already formed as assessed by ultrasound imaging of the lungs of live mice and **H&E staining H&E analysis** of the mouse lung sections before treatment (data below and **Supplementary Fig. 10**).

H&E staining of orthotopic LLC tumors at 12 days after transplantation, before initiation of treatment

Representative ultrasound images of orthotopic LLC tumors at 12 days after transplantation, before initiation of treatment

In Fig. 6c (**Fig. 6e, f** in revised version) treatment of mice with ISRIB was initiated 10 weeks after KRAS G12D induction by intratracheal intubation of CRE lentivirus. At that time of tumor progression, the mice had formed tumors readily detectable by ultrasound imaging (data below and **Supplementary Fig. 11**).

Representative ultrasound images of KRAS G12D lung tumors in mice at 10 weeks after intratracheal intubation of CRE-lentivirus

In a new experiment we show that ISRIB treatment of mice bearing sizable (200 mm³) subcutaneous KRAS G12D tumors resulted in substantial inhibition of tumor growth (data below and **Supplementary Fig. 11**).

The above data show that ISR inhibition exhibits anti-tumor effects on formed lung tumors.

Question#3. *The in vivo data demonstrating the efficacy for ISRIB treatment and PERK inhibitor presented in figure 6 is striking but is not accompanied by data demonstrating on-target efficacy. Can the authors demonstrate on-target effects, is phospho-eIF2a reduced in these tumors? In this scenario it would also be helpful to assess if there are also differences in phospho-ERK and DUSP6 expression.*

ANSWER: We have examined ATF4, DUSP6 and phospho-ERK levels in LLC lung tumors treated with vehicle control or ISRIB (**Fig. 6d** of revised version). The data showed a decrease in ATF4 expression, which is as a marker of ISRIB treatment efficacy. ISRIB treatment increased the expression of DUSP6 and decreased the phosphorylation of ERK in LLC tumors compared to tumors treated with vehicle/control (n= tumor number).

Question#4. *GSK2606414 is described as “a potent and selective PERK inhibitor”. This 2017 study (PMC5442476) found GSK2606414 in addition to reducing PERK activity was a potent RIPK1 inhibitor suggesting this inhibitor is not selective to PERK. Based on this finding with GSK2606414 should be replicated using additional selective approaches such as genetic manipulation of PERK expression.*

ANSWER: We have used a genetic approach to verify the implication of PERK in mutant KRAS lung tumor formation. Specifically, we found that PERK downregulation by a pool of 4 different siRNAs impaired the survival of phospho-eIF2 proficient (S/S) KRAS G12D cells but had no effect on the survival of phospho-eIF2-deficient (A/A) cells (see colony formation assays below and **Fig. 4a, b** of revised version). Downregulation of PERK by siRNAs partially decreased phospho-eIF2 in S/S tumor cells compared to A/A tumor cells due to the presence of the other 3 eIF2 kinases in the cells (see blots below and **Fig. 4a** of revised version). Considering that PERK downregulation selectively reduces the colony formation efficacy of S/S tumor cells to equal levels of colony formation efficacy of A/A cells indicates the major role of PERK in KRAS G12D tumor cell survival through phospho-eIF2 (see graphs below). These data are in complete agreement with those obtained from the treatments of KRAS G12D S/S and A/A tumor cells with the PERK inhibitor GSK2606414 in **Fig. 4c** of revised version. The modest but statistically significant increase of A/A cells treated with PERK siRNAs indicates that implication of PERK in pathways independent of phospho-eIF2. This is further explained in the responses to specific comments below.

Because GSK2606414 can inhibit RIPK1 in addition to PERK (PMC5442476), we further tested the effects of the potent RIPK1 inhibitor GSK963 on the survival of human LUAD cells. The data below show that RIPK1 inhibition did not affect the survival of LUAD cells bearing KRAS G12C

(H23 and H358 cells) as opposed to PERK inhibition with GSK2606414, which impaired the survival of these tumor cell lines (**Supplementary Fig. 7** of revised version).

Question #5. A recent study by Unni AM et al (eLIFE 2018) suggests that *DUSP6 upregulation is required in LUAD to ensure ERK activity stays within a certain threshold. They demonstrate a reduction in DUSP6 levels associates with an increase in p-ERK and cell death. In this study the authors conclusions are slightly different they suggest a decrease in DUSP6 expression mediated via the ISR is required for efficient tumor growth. It would be good to place the findings of this study in context with the previously published literature in the discussion.*

ANSWER: The eLIFE 2018 article nicely showed the therapeutic potential of the hyperactivation of phospho-ERK in response to treatment of human LUADs with the DUSP inhibitor BCI. The eLIFE 2018 study is in line with previous works demonstrating the anti-proliferative/pro-senescent effects of hyperactivated MAPK signaling in tumors (see review in Cancer Res. 2014; 74: 412-419).

To better address this matter in our study, we tested the implication of DUSP6 in the survival/proliferation of KRAS G12D tumor cells by genetic means (see data below, which are **Fig. 3e, f** of revised version). We found that DUSP6 downregulation by a pool of 4 different siRNAs stimulated ERK phosphorylation in phospho-eIF2-deficient (A/A) KRAS G12D cells to equal levels of phosphorylated ERK in phospho-eIF2-proficient (S/S) KRAS G12D cells treated with scrambled siRNAs. Moreover, DUSP6 downregulation increased the survival of KRAS G12D (A/A) cells to equal levels of survival of S/S cells treated with scrambled siRNAs. On the other hand, DUSP6 downregulation increased phospho-ERK and modestly decreased the survival of phospho-eIF2-proficient (S/S) KRAS G12D cells although the latter effect was not statistically significant (3 biological replicates performed in triplicates). *Thus, stimulation of phospho-ERK in KRAS G12D S/S cells by DUSP6 suppression does not induce anti-proliferative effects.* That is, stimulation of phospho-ERK by phospho-eIF2 is below the threshold required to induce anti-proliferative effects in KRAS G12D lung tumor cells. It should also be noted that the pharmacological inhibitor BCI used in the eLIFE 2018 study targets DUSP1 and 6 (Chembiochem. 2014;15:1436-45), both of which dephosphorylate ERK. Such a broad specificity of BCI could

account for the inability of DUSP6 downregulation alone to increase phospho-ERK at levels that can elicit strong anti-proliferative effects in KRAS G12D S/S cells.

Specific comments

Figure 2. Is the RNA seq data set available? I didn't notice a statement pertaining to data availability in the manuscript.

ANSWER: The GEO accession number is: GSE155238. The secure token is **mroxmygkzpsbmd**. This information is now included in the manuscript.

Figure 3b. A blot demonstrating a functional inhibition of the ISR e.g. ATF4 would add to this figure panel. Also, an explanation of why serum deprivation conditions were used would be useful in the text. Is it to activate the UPR hence increase PERK signaling?

ANSWER: We have included ATF4 levels in **Fig. 3b** of revised version (see blot below). Previous studies employed serum stimulation to obtain robust effects on stimulation of MAPK signaling in mouse KRAS G12D lung tumor cells (Cancer Cell 2004; 5:375-87; Cancer Cell 2012; 22:222-34).

Figure 4a. Blots verifying the functional effect of GSK2606414 should be included to demonstrate a reduction in PERK phosphorylation and a reduction in eIF2 α -p in eIF2 α S/S. Given the issues surrounding the selectivity of GSK2606414 an additional approach e.g. knockdown of PERK via siRNA or shRNA and assessment of the outcome on DUSP6, p-EIF2 α and p-ERK should be included. Is the decrease in colony formation a result of increased cell death?

ANSWER: We tested the effects of the PERK inhibitor GSK2606414 in mouse KRAS G12D cells by immunoblotting (see below and **Fig. 4d** of revised version).

As explained in General Comments **Question#4** above, our new data with *PERK* siRNAs showed that GSK2606414 acts as a *PERK* inhibitor in mouse and human LUADs. This data is now shown in **Fig. 4a, b**.

We have tested the effects of GSK2606414 on the mouse KRAS G12D cell death by FACS. The data below show that the *PERK* inhibitor (*PERKi*) selectively stimulates the death (i.e. propidium iodide stained cells in subG₁ fraction) of phospho-eIF2 proficient (S/S) but not of deficient (A/A) KRAS G12D cells. This data further supports our conclusion that *PERK* is the major eIF2 kinase to promote the survival of mutant KRAS lung tumor cells (now shown in **Supplementary Fig.6**).

Furthermore, we examined the levels of cleaved caspase 3 and Ki-67 in mouse LLC tumors from Fig. 6b (original version) treated with ISRIB (see below; it is now **Supplementary Fig. 10b**). We found that treatment with ISRIB significantly increased the expression of cleaved caspase 3 and decreased Ki67 consistent with anti-apoptotic and anti-growth effects of the ISR inhibitor.

Figure 4b. There is basal activation of PERK in the eIF2a S/S cells as demonstrated by the detection of p-PERK and p-eIF2a. What is the outcome if eIF2a S/S cells are treated with PERKi without the addition of TG? Do you see an increase in DUSP6 levels and a decrease in p-eIF2a?

ANSWER: We have performed the immunoblotting analysis of mouse KRAS G12D S/S and A/A cells side by side (see blot below). The data below show no significant differences in basal PERK activity (PERK-P) between the two cell types. This data is now shown in **Fig. 4d**.

The effects of PERK inhibitor in the absence of TG treatment are shown in the blot below. There is no detectable difference in phospho-ERK levels between untreated KRAS G12D S/S cells and cells treated with the PERK inhibitor only. The data indicated the need of stress to detect the effects of PERK inhibition on phospho-eIF2 and phospho-ERK in cultured cells.

Figure 5a. Similar to my comment above what is the outcome on eIF2α, DUSP6 and ERK in eIF2α S/S cells if you add ISRIB alone and not in combination with thapsigargin (Tg)? I understand that addition of Tg is a way to “boost” PERK activation, but it may not accurately model events in vivo. The level of PERK activation instigated by Tg could be many folds higher than that achieved in vivo, in addition Tg treatment may activate additional cellular pathways not triggered in vivo.

ANSWER: The immunoblot below shows that ISRIB does not inhibit ERK phosphorylation nor does it upregulate DUSP6 in eIF2α S/S in the absence of stress. Several studies have thoroughly characterized ISRIB action in cultured cells and showed that this compound does not elicit profound effects on cell proliferation and general mRNA translation unless it is combined with ER stressors (Elife. 2013;2:e00498; Elife. 2015;4: e05033 ; Elife. 2015;4:e07314).

Figure 5a (cont'd) It is also interesting to note that the eIF2a A/A cells have endogenously high levels of ATF4 do the authors have any thoughts on why this is the case? If an ISR is not functional in this setting it seems surprising to see high ATF4.

ANSWER: The immunoblot data shown in our response to the Reviewer's comments on Figure 3b as explained above shows that ATF4 levels are indeed lower in phospho-eIF2 deficient (A/A) cells compared to proficient S/S cells. The data in Fig. 5a of original version, which the Reviewer alluded to, were obtained from the analysis of ISR signaling in ISRIB-treated S/S and A/A cells on separate blots for each cell type. We will include new data for **Fig.5a** obtained from the analysis of ISR signaling in S/S and A/A cells on the same blots. The data verify the lower ATF4 levels in A/A than S/S cells.

Figure 6. Connecting the proposed mechanism to the functional in vivo observations would really strengthen this study. While the effect of ISRIB on tumor growth is clear whether this is achieved via the proposed pathway is less developed. Evidence of on-target effect of the inhibitors in Fig 6 should be provided i.e. p-eIF2a blots and evidence of ISR activation (ATF4), DUSP6 expression changes and p-ERK expression in tumors from this in vivo experiment would be a good starting point.

ANSWER: As explained above in General Comments, Question#3, we have tested the effects of ISRIB on ATF4, DUSP6 and phospho-ERK in LLC lung tumors grown in mice by IHC analysis as shown below. Downregulation of ATF4 served as a marker of ISRIB efficacy for tumor treatment. The data indicate the on-target effects of ISRIB on DUSP6 upregulation and phospho-ERK inhibition in treated tumors (the data shown in Fig. 6d).

If BCI is used in vivo do you observe a reduction in tumor growth similar to that seen with ISRIB or PERK inhibitor? Do knockdown DUSP6 LUAD cells form tumors more rapidly?

ANSWER: BCI is an inhibitor of DUSP1 and 6 (Chembiochem. 2014;15:1436-45). As explained in our response to General Comment Question#5 above, we tested the implication of DUSP6 in the survival/proliferation of KRAS G12D tumor cells by genetic means (siRNAs). *The data clearly supported an anti-proliferative role of upregulated DUSP6 in mouse KRAS G12D eIF2 (A/A) cells.*

In relation to the efficacy of GSK2606414 in Fig 6 H1299 WTKRAS do the authors have any thoughts why GSK2606414 would reduce tumor growth in this setting and ISRIB would not?

ANSWER: The reduction of H1299 WT KRAS tumor growth by the PERK inhibitor GSK2606414 in Fig. 6a is not statistically significant ($p=0.116$). Nevertheless, this trend can be explained by the implication of PERK in tumorigenic pathways independent of p-eIF2 like, for example, the NRF2 pathway in pancreatic and lung cancers (Cell Death Dis. 2021; 12:82). It remains possible that the NRF2 pathways is sensitive to PERK inhibition in WT KRAS cells. On the other hand, ISRIB acts downstream of PERK at the level of p-eIF2, and therefore its effects are exerted through the antagonism of eIF2-dependent translation, which results in the upregulation of DUSP6 and inhibition of p-ERK in mutant KRAS but not WT KRAS cells (Fig. 5). This is now explained in the Discussion of the revised manuscript.

As stated in the general comments it would be interesting to model a scenario similar to that observed in LUAD patients where tumors are not detected until a more advanced stage. What is the outcome on tumor growth if tumors are allowed to firstly establish to various sizes and then treated with PERKi or ISRIB?

ANSWER: As explained in our response to General Comment Question#2 above, initiation of treatments with ISR inhibitors took place when the formation of lung tumors was readily detectable by IHC and ultrasound imaging. This is better described in the revised manuscript and supported by data in **Supplementary Figs. 10, 11**.

REVIEWER #2 (Remarks to the Author):

Question#1. *The role of DUSP6 as a downstream effector of ISR is currently correlative. a. Does suppression of DUSP6 expression (shRNA, sgRNA or expression of dominant negative) in eIF2 α /A cells restore cell proliferation? b. Does suppression of DUSP6 expression (shRNA, sgRNA or expression of dominant negative) overcome the effect of a PERK inhibitor in colony formation assays?*

ANSWER: (a, b) We tested the implication of DUSP6 in the survival/proliferation of KRAS G12D tumor cells by genetic means (see data below; **Fig. 3e, f** in revised version). We found that DUSP6 downregulation by a pool of 4 different siRNAs stimulated phospho-ERK in KRAS G12D eIF2 α A/A cells to equal levels of phospho-ERK in KRAS G12D eIF2 α S/S cells. Also, DUSP6 downregulation increased the survival of KRAS G12D eIF2 α A/A cells to equal levels of the survival of eIF2 α S/S cells treated with scrambled siRNAs. We noticed that DUSP6 downregulation modestly decreased the survival of KRAS G12D eIF2 α S/S cells although this effect was not statistically significant (3 biological replicates performed in triplicates). This effect possibly indicates that ~~that~~ hyperactivation of p-ERK by the downregulation of DUSP6 in KRAS G12D eIF2 α S/S cells has the potential to inhibit proliferation. This is consistent with a study in *eLife* 7, e33718 showing that hyper-activation of p-ERK by the pharmacological inhibition of DUSPs can inhibit the growth of human LUAD cells. However, our data show that threshold required for p-ERK to induce anti-proliferative effects cannot be reached by the downregulation of DUSP6 in KRAS G12D eIF2 α A/A cell. *Thus, the new data support our original conclusion that DUSP6 acts as tumor suppressor and its upregulation accounts for the growth inhibition of phospho-eIF2 deficient (A/A) lung tumor cells.*

Question#1 (cont'd) c. *Is DUSP6 expression increased in tumors from ISRIB-treated mice?*

ANSWER: We performed IHC assays to assess ATF4 (marker of ISRIB effectiveness), DUSP6 and phospho-ERK levels in ISRIB-treated LLC tumors grown in immunocompetent mice (**Fig. 6c, d** of revised version). The data below show the downregulation of ATF4 indicating of tumor

response to ISRIB treatment. ISRIB-treated LLC tumors contained decreased levels of phospho-ERK, and increased DUSP6 protein expression compared to vehicle/control-treated tumors (n= tumor number). This data is included in **Fig. 6d** of revised version.

Question#1 (cont'd) d. Does ISRIB treatment in the absence of thapsigargin (TG) have any impact of DUSP6 expression in eIF2a^{S/S} cells?

ANSWER: The immunoblot below shows that ISRIB does not elicit profound effects on DUSP6 expression in eIF2a^{S/S} in the absence of stress. Several studies have thoroughly characterized ISRIB action in cultured cells and showed that this ISR inhibitor does not influence cell proliferation and general mRNA translation unless it is combined with ER stressors (Elife. 2013;2:e00498; Elife. 2015;4: e05033 ; Elife. 2015;4:e07314).

Question#2 The conclusion that ISR inhibition is effective in mutant KRAS and not wildtype KRAS cells is not sufficiently supported by experiments performed in a single KRAS WT cancer cell line H1299. Although H1299 is wildtype for KRAS, it has a missense mutation in NRAS. It is surprising that a presumably general stress response pathway in cancer cells is specific to KRAS mutant cells but not that of NRAS. Investigating this in additional KRAS WT cell lines such as HBEC3, and additional NRAS mutant cell lines such as H2087 would further support the conclusion.

ANSWER: The colony formation assays in **Suppl. Fig. 7** of the revised version show that pharmacological inhibition of PERK more highly decreased the survival of H23 and H358 cells with KRAS G12C than H1299 and H1703 cells with wild type KRAS. Although H1299 cells contain NRAS mutation, H1703 cells do not. As such, we employed the lung cancer **H1703 cells**, which contain WT alleles of all RAS isoforms, to overexpress either GFP-wild type KRAS 4B or GFP-KRAS 4B G12C by retrovirus infection (the suitability of GFP-KRAS constructs was demonstrated in *Nat Commun.* 2017;8:15205). We found that GFP KRAS G12C overexpression stimulated the PERK/p-eIF2 arm and p-ERK compared to WT KRAS overexpression (see blot below). This data is included in **Supplementary Fig. 6b**.

Concerning HBEC3 cells, we noticed from published studies that mutant KRAS (G12D or G12V) overexpression in these cells did not increase MAPK signaling nor did it render them tumorigenic in immune deficient mice (Sato et al. *Cancer Res.* 2006; 66:2116-28 and Ihle et al. *J Natl Cancer Inst* 2012:104:228-239). Because the effects of p-eIF2 are exerted at DUSP6/p-ERK level, we reasoned that HBEC3 is not a suitable system to address the role of ISR in MAPK signaling in mutant KRAS lung cancers.

We further examined ISR signaling in H1299 cells overexpressing either WT KRAS or mutant KRAS (G12C, G12D or G12V) previously established and characterized in *Ann Oncol* 2011;22:

235-237, *Oncotarget* 2015; 6:30072-30087 and *Scientific Reports* 2016; 6:28398. We observed a consistent pattern of upregulation of PERK/p-eIF2 arm and p-ERK in the tumor cells overexpressing the mutant KRAS proteins compared to isogenic cells overexpressing WT KRAS (see blot below). Upregulation of ISR signaling in H1299 overexpressing the mutant KRAS proteins was evident in the absence of TG and further enhanced after TG treatment for 1h (data below; **Supplementary Fig. 6b** of revised version). Thus, mutant KRAS overexpression elicited similar effects on ISR signaling and p-ERK in H1703 (data above) and H1299 cells (data below).

We further compared the colony forming efficacy of H1299 and H1703 cells overexpressing either WT KRAS or KRAS G12C in response to PERK inhibitor treatment. The data below show that KRAS G12C overexpression rendered both lung tumor cell lines increasingly susceptible to PERK inhibition compared to their isogenic counterparts overexpressing WT KRAS (shown in **Supplementary Fig. 7b**). Collectively these findings further supported the interpretation that *NRAS* mutation in H1299 cells does not interfere with the effects of mutant KRAS overexpression on ISR signaling and tumor cell survival.

Question#3. *In vivo study: a. Is the effect of ISRIB in vivo cytotoxic or cytostatic? b. Please include a marker to validate ISRIB effect in vivo, e.g. ATF4 IHC in vehicle vs ISRIB treated tumors.*

ANSWER: a. IHC analysis indicated a higher staining of ISRIB-treated tumors for cleaved Caspase 3 and lower staining of Ki-67 than tumors treated with vehicle control in tumors from figure 6b (original version) (n= tumor number). The data below is shown in **Supplementary Fig. 10b**.

Mouse LLC tumors grown in syngeneic mice treated with ISRIB

b. As explained in our response to Question #1 above, ATF4 expression was downregulated in mouse LLC tumors treated with ISRIB (see data in response to Question#1).

Other concerns:

Question#4 *If possible, please show better phospho-ERK IHC image for Figure 3a. The signal looks very weak in general. An image like the one presented in FigureS3c shows phospho-ERK expression much more clearly.*

ANSWER: We have obtained better images of phosphorylated ERK in tumors shown in **Fig. 3a**.

Question#5 *DUSP6 IHC in Figure 3e looks nuclear, can the authors comment on this since DUSP6 is a cytoplasmic protein?*

ANSWER: Better images now in **Fig. 3d** show cytoplasmic staining of DUSP6. This new experiment was performed with a higher dilution of the anti-DUSP6 antibody. The anti-DUSP6 antibody was used at a dilution of 1:100 in the figure below as opposed to 1:50 in the original version.

Question#6 *Figure 4b and c should be run on the same gel to provide direct comparison of the effect of PERKi on S/S cells to A/A cells at baseline.*

ANSWER: We have repeated this blot and the data is shown below. This new data is shown in **Fig. 4d**.

Question#7 *“Normal” KRAS cells was used throughout the manuscript. “Wildtype” should be used instead.*

ANSWER: We corrected throughout the manuscript.

Question#8 *Supplementary figure 4: a positive relationship is not evident at all based on the data presented. It is not clear if this adds much value to the manuscript.*

ANSWER: Suppl Fig 4 (original version) shows 4 scatterplots with overlaid generalized additive models (GAM). All four tumors show highly significant weak-to-moderate positive associations between cytoplasmic phospho-eIF2 levels and nuclear phospho-ERK as assessed by Spearman's rank correlation. The smoothed GAMs indicate monotonic relationships in tumors 2 and 3. In tumors 1 and 4 there are negative relationships seen in the GAMs in the less confidently assigned sparsely populated areas of the plots, but the well-populated regions are clearly positively related.

REVIEWER #3 (Remarks to the Author):

Major comments:

Question#1. *The analysis performed on the LUAD TMA shows a clear correlation of high p-eIF2 α levels with the more aggressive histological subtypes, as well as with tumors with high invasive and proliferative markers. Yet, given the results presented later in the manuscript in my opinion it is essential to analyze the survival differences in high/low p-eIF2 α patients harboring different driver oncogenes. This would be particularly important in KRAS mutant patients given the specificity of the ISR inhibition and therapeutic efficacy in this subtype suggested in the manuscript.*

ANSWER: We have examined the prognostic value of phospho-eIF2 in the survival of patient subgroups with oncogenic mutations in the most common drivers (KRAS or EGFR) or P53. The data below reveal a trend towards poor survival with high phospho-eIF2 after surgery, but the difference loses significance in this subgroup analysis, probably due to small sample size. Overall, the data in **Fig. 1, Supplementary Fig. 1** and **Table 1** suggest that high phospho-eIF2 determines poor outcome in all contexts, but it will certainly be valuable to examine the importance of genomic context in our future work.

It is noteworthy that the KRAS mutation status is not the only oncogenic determinant of mutant KRAS cancers. Specifically, Yuan et al., Cell Reports 2018; 22:1889-1902 nicely showed the presence of two cellular contexts that

define mutant KRAS cancers. That is, (i) tumors that maintain addiction to mutant KRAS and maximally activate MAPK signaling and (ii) tumors in which oncogene addiction switched from mutant KRAS to the RSK-mTOR pathway. It is not possible from the DNA seq data of the human LUAD specimens to determine which of the two contexts are associated with increased phospho-eIF2. The implication of phospho-eIF2 in the upregulation of MAPK pathway via DUSP6 from our study suggests the tumorigenic function of ISR in tumors addicted to mutant KRAS. Presently, it is unknown whether phospho-eIF2 levels can determine a switch of mutant KRAS-addicted to RSK-mTOR-addicted tumors. However, previous findings from our laboratory and collaborators have shown an inverse relationship between phospho-eIF2 and mTOR signaling (Mol Cancer Res 2015; 13:1377-1388 and Nat Commun. 2016; 7:11127), which implies a potential role of ISR in the switch of the two contexts of mutant KRAS LUADs. This is a matter of future investigation.

Question#2. As a follow up to point #1, conceptually, I have nothing against having selected KRAS-mutant LUAD to evaluate the therapeutic efficacy of ISR inhibitors. KRAS mutations define an aggressive disease with few therapeutic options. Yet I do not understand the mechanistic basis for its KRAS specificity as suggested within the manuscript. The authors claim several times throughout the text (as an example see line 190, data support the specificity of ISRIB in targeting p-eIF2 α function in mutant KRAS cells).

ANSWER: We have obtained additional data shown below and in **Supplementary Fig. 6** showing that mutant KRAS increases PERK/p-eIF2 signaling and sensitivity of LUAD cells to PERK inhibition. Specifically, we show below that overexpression of GFP-KRAS G12C in H1703 lung cancer cells. ~~Which~~ cells, which contain WT KRAS, upregulates PERK/p-eIF2 and p-ERK compared to the same cells overexpressing GFP WT KRAS. Also, overexpression of different forms of mutant KRAS (G12C, G12D or G12V) in H1299 cells with endogenous WT KRAS exhibits stimulatory effects on PERK/p-eIF2 compared to the same cells overexpressing WT KRAS. Moreover, comparison of different lung tumor cell lines with either wild type KRAS (H1703) or KRAS G12C (H23, H258) indicated the upregulation of PERK/p-eIF2 and p-ERK in mutant KRAS cells compared to WT KRAS cells.

Furthermore, colony formation assays of human LUADs, which are shown below and in **Supplementary Fig. 7**, supported a higher susceptibility of mutant KRAS cells to ISR inhibition than wild type KRAS. These data suggested that mutant KRAS upregulates ISR and increases the susceptibility of LUAD tumors to ISR inhibition.

Question#2 (cont'd) Data presented in the manuscript include Gene set enrichment analyses suggesting the ^{ISEP} involvement of the p-eIF2 α -dependent genes in the stimulation of growth factor receptor signaling (Supplementary Fig.7). I assume, this would also apply to EGFR-mutant tumors.

ANSWER: We have tested the effects of phospho-eIF2 on PTEN expression and phosphorylated AKT as markers of Growth Factor Receptor Signaling as indicated by Gene set enrichment analyses (Supplementary Fig. 12 of revised version). We have not observed significant differences in the regulation of PTEN and phospho-AKT between proficient (S/S) and phospho-eIF2-deficient (A/A) tumors (see data below). We have not investigated the implication of ISR in EGFR signaling in human LUAD cells. The data below further support our interpretation that stimulation of p-ERK via the downregulation of DUSP6 represents a major mechanism of p-eIF2 action in mutant KRAS cells.

Question#3 Even more confounding are the experiments using the H1299 cells, both in vitro (Figure 5 & Supp Figure 5) and upon implantation in nude mice (Figure 6). H1299 are NRAS-mutant cells (Gln61Lys) and should, presumably, function through the same signaling pathways as the KRAS counterparts. This reviewer wonders if the differential response observed could be due to the overall stress present in these cell lines.

ANSWER: We now present new data (below and **Supplementary Fig. 6**) showing the upregulation of ISR pathway by mutant KRAS in human LUAD cells. Specifically, the data show that overexpression of different mutant KRAS proteins (G12C, G12D or G12V) in H1299 cells stimulates the p-PERK, p-eIF2 and p-ERK compared to the same cells overexpressing WT KRAS. These effects are detected in unstimulated cells and become more evident after stimulation with the ER stressor thapsigargin (TG). The data suggested that the effects of overexpressed mutant KRAS proteins are dominant over endogenous mutant NRAS for the stimulation of ISR signaling and p-ERK in H1299 cells.

To exclude side effects of mutant NRAS on ISR signaling in mutant KRAS cells, we employed H1703 lung tumor cells containing WT alleles of all the RAS isoforms. We observed that overexpression of GFP-KRAS G12C in H1703 cells upregulated the PERK/p-eIF2 arm and p-ERK compared to the same cells overexpressing GFP-WT KRAS (data below; **Supplementary Fig. 6c**). Thus, KRAS G12C overexpression upregulates ISR and p-ERK regardless of NRAS mutation.

Moreover, colony formation assays indicated that both H1299 and H1703 cells overexpressing KRAS G12C were more susceptible to treatments with the PERK inhibitor GSK2606414 than their isogenic counterpart cells overexpressing WT KRAS (see below and **Supplementary Fig. 9b**).

Therefore, H1299 and H1703 respond similarly to ISR inhibition despite the expression of mutant NRAS in H1299 cells. Collectively, the findings support the interpretation that mutant KRAS stimulates ISR signaling and sensitizes lung tumor cells to ISR inhibition.

Question#3 (cont'd) *Within the text the authors refer a stress phenotype, caused by increased DNA damage, proteotoxic, metabolic and oxidative stress because of oncogenic mutations. Could the authors analyze markers for some of these stress forms to evaluate a putative correlation to the response to ISR inhibitors?*

ANSWER: Mutant RAS increases oxidative, proteotoxic and genotoxic stress (Cancer Cell 2011;20:281-282; Cancer Res. 2019; 79:5849-5859; Cancer Cell 2014; 25:243-256; Cell Reports 2018; 22:1889-1902). These stress forms are inducers of ISR signaling in different models of human disease including cancer (Science. 2020;368:6489). We have partially addressed the connection between KRAS, p-eIF2 and oxidative stress in lung tumor cells. We used H1703 cells overexpressing either GFP-WT KRAS or GFP-KRAS G12C, because mutant KRAS overexpression in these cells upregulates ISR signaling (data above and **Supplementary Fig. 6c**). Treatment of H1703 cells with hydrogen peroxide caused a higher ROS production in KRAS G12C cells compared to their WT KRAS isogenic counterparts (see data below). This is in line with previous work demonstrating a higher ROS production in mutant KRAS tumor cells subjected to pro-oxidant treatments compared to isogenic cells with WT KRAS (Proc Natl Acad Sci U S A. 2011;108:8773-8). The same study showed that high ROS production rendered mutant KRAS tumor cells increasingly susceptible to pro-oxidant anti-tumor therapies compared to isogenic cells with WT KRAS (Proc Natl Acad Sci U S A. 2011;108:8773-8). The data below provided evidence for a positive connection between ROS upregulation and increased ISR signaling in mutant KRAS cells.

We used oxidative stress as a reference point to inquire about the implication of ISR in stress responses induced by mutant KRAS in the mouse KRAS G12D lung tumor cells. As shown below, we found a higher production of ROS in p-eIF2-deficient (A/A) cells after treatment with hydrogen peroxide compared to p-eIF2 (S/S)-proficient cells. *These data supported an antioxidant function of p-eIF2, which is linked to its ability to promote the survival and growth of mutant KRAS G12D tumor cells.* A source of oxidative stress in mouse KRAS G12D lung tumors may be caused by the ability of p-eIF2 α to disrupt mitochondrial respiration and oxidative phosphorylation as indicated by GSEA analysis in **Supplementary Fig. 12** of revised version. The antioxidant function of p-eIF2 in mutant KRAS tumor cells may also indicate a therapeutic potential of ISR inhibitors in combined treatments with pro-oxidant therapies.

Furthermore, we obtained preliminary evidence that mutant KRAS causes metabolic stress as indicated by the increased phosphorylation of AMPK α in H1703 cells expressing GFP-KRAS G12C compared to isogenic cells with GFP-wild type KRAS (see data below). Similarly, phospho-

AMPK α is upregulated in KRAS G12C-overexpressing compared to WT KRAS-overexpressing H1299 tumors isolated from mice (see data below). Considering the upregulation of ISR by mutant KRAS (**Supplementary Fig. 6**), ROS and metabolic stress (increased AMPK α phosphorylation) may account, at least in part, for stimulation of ISR in mutant KRAS lung tumors. Because p-AMPK α is a marker of stress from energy starvation and increased LKB1 activity, ISR signaling may also be affected by the status of LKB in lung tumors. It is conceivable that the connection between oxidative and metabolic stress with ISR signaling in lung cancer is a vast field of investigation which cannot be addressed in a single study. Nevertheless, our study provides the basis for future investigation of this interesting research topic.

Question#3 (cont'd) Do H23 and H358 cells (used in Figure 5) display higher stress markers than the non-responsive H1299? Similarly, is this also the case in sections from implanted H1299 cells overexpressing exogenous KRAS12C (Figure 6) when compared to the controls overexpressing the KRAS wild type?

ANSWER: The data above with ROS and AMPK α phosphorylation supported the implication of stress caused by mutant KRAS in the upregulation of ISR signaling. In addition, we have compared ISR signaling between H23 (KRAS G12C), H358 (KRAS G12C), H1703 cells (WT KRAS) prior to and after stimulation with thapsigargin (TG; positive control). The data below (now included in **Supplementary Fig. 6**) show that markers of ISR signaling, namely p-PERK, p-eIF2 and ATF4, are upregulated in mutant KRAS cells (H23, H358) compared to WT KRAS containing cells (H1703). Also, ISR signaling and p-ERK are both upregulated in H1299 cells overexpressing mutant KRAS (G12C, G12D, G12V) compared to H1299 cells overexpressing WT KRAS. Furthermore, upregulation of ISR signaling and p-ERK are evident in H1703 cells overexpressing GFP-KRAS G12C compared to the same cells overexpressing GFP-WT KRAS. Collectively, these data support the conclusion that mutant KRAS is a stimulator of ISR signaling. Because mutant KRAS is an inducer of many forms of stress as explained above, *most probably ISR integrates multiple stress pathways to promote its tumorigenic effects through p-eIF2 α in mutant KRAS lung cancers.* The characterization of each stress form in the stimulation of ISR in mutant KRAS lung cancer requires many and lengthy studies, which will be greatly facilitated by the tools developed in this study.

Question#4. *The chemically induced carcinogenic experiment was performed on haploinsufficient eIF2αS/A mice. Does this mean that partial eIF2α inhibition could be therapeutically relevant in the clinic? Or does this imply some sort of dominant-negative effect of the S51A mutant?*

ANSWER: The data with urethane treated mice bearing a heterozygous germline eIF2αS/A mutant allele (Supplementary Fig. 3 of revised version) suggested that 50% reduction of phospho-eIF2 can inhibit mutant KRAS lung tumorigenesis. The mutant eIF2αS/A allele in these mice exhibits a haploinsufficient phenotype and does not act as dominant negative to endogenous WT allele (Mol Cell. 2001;7:1165-76).

Question#5. *The abstract indicates (line 32) “phospho-eIF2α causes specific translational repression of dual specificity phosphatase 6”. I am not sure what the authors mean by specific since the transcriptional variations are pleiotropic.*

ANSWER: The term “specific” refers to translation suppressive function of phospho-eIF2 on DUSP6 expression. We reached this conclusion by analyzing the translatability of DUSP6 mRNA in polysome profile assays of mouse KRAS G12D tumor cells that were either proficient (S/S) or deficient in p-eIF2 (A/A). Our data in Fig. 3c showed the reduction of DUSP6 mRNAs bound to polyribosomes of S/S cells compared to DUSP6 mRNAs bound to polyribosomes of A/A cells after normalization to total DUSP6 mRNAs expressed in S/S and A/A cells, respectively. In these assays, DUSP6 mRNAs were further normalized to endogenous glyceraldehyde 3-phosphate dehydrogenase (GAPDH) and ACTIN mRNAs in the total and poly-ribosomal fractions of S/S and A/A cells. This approach determined the specificity of p-eIF2α in DUSP6 mRNA translation after normalization to background translational effects of phospho-eIF2α on housekeeping mRNAs. These conditions are explained in the manuscript text and legend of Fig. 3 of the revised version.

Question#5 (cont'd). *For instance, the PTEN phosphatase is significantly down-regulated in eIF2α-impaired cell lines as indicated in Figure 2g. Is the activity of the PI3K-Akt pathway*

increased? This could be assessed in the serum starvation and release (Figure 3b) as well as in sections from eIF2 α /A/A tumors (Figure 3e). The PI3K pathway is also essential an essential pathway in KRAS-driven LUAD as demonstrated by work from the Downward laboratory and others.

ANSWER: We have tested PTEN expression and AKT phosphorylation in mouse KRAS G12D S/S and A/A tumor cells (see data below). The data showed the lack of significant effects of phospho-eIF2 on both proteins in mouse KRAS G12D cells. These data support the conclusion of our study that stimulation of phospho-ERK via the downregulation of DUSP6 represents a major mechanism of phospho-eIF2 action in mutant KRAS cells.

Question#6. *The potential use of compounds targeting the ISR is based on the in vivo experiments shown in Figure 6c. Yet, whether these results could be extended to patients is unclear. The drug treatment was initiated 10 weeks after tumor initiation by lentiviral infection. It will be essential to characterize what is the tumor grading at the beginning of the treatment and if there are advanced lesions at this stage. Alternatively, do the authors have data from treatments started later point when the mice are present with advanced disease?*

ANSWER: We now better clarify the conditions of tumor treatments with ISR inhibitors in **Fig. 6**. Treatments of orthotopic mouse LLC tumors with ISR inhibitors in **Fig. 6c, d** were initiated after the detection of sizable tumors by H&E staining, IHC analysis and ultrasound imaging (see images below and data in **Supplementary 10a**).

H&E staining of orthotopic LLC tumors at 12 days after transplantation, before initiation of drug treatment

Representative ultrasound images of orthotopic LLC tumors at 12 days after transplantation, before initiation of treatment

Treatment of KRAS G12D lung tumors in mice with ISRIB (**Fig. 6e, f**) was initiated at 10 weeks after tumor induction by the lentivirus delivery of CRE (**Fig. 6e**). At that time point, ultrasound imaging indicated the formation of lung tumors (data below and in **Supplementary Fig. 11b**).

**Representative ultrasound images of KRAS G12D lung tumors in mice
at 10 weeks after intratracheal intubation of CRE-lentiviruses**

Question#7 Likewise, to what extent this experiment with endogenous tumors (Figure 6c) mirrors the clinical scenario is unclear in another point. The rationale within the manuscript up to this point was the comparison between high vs low eIF2 α LUAD. To this extent the authors used a transgenic strain to overexpress this factor in the context of a KRAS12D oncogene and compared that to the control strain harboring a non-phosphorylatable and therefore inactive eIF2 α S51A allele. This comparison clearly demonstrated that eIF2 α overexpression induces a more aggressive disease. Yet the experiment shown in Figure 6c is based exclusively on the inducible KRas12D strain and thereby these tumors would express “normal” eIF2 α level

ANSWER: We would like to clarify that the way the p-eIF2 knock in mouse was developed necessitated the comparison KRASG12D; fTg eIF2 α S/S and KRASG12D; fTg eIF2 α A/A lung tumors (Cell Metab. 2009;10(1):13-26). The mouse KRAS G12D strain, which exclusively harbors endogenous eIF2, wouldn't have been the proper control in experiments with mice deficient in phospho-eIF2 (namely KRASG12D; fTg eIF2 α A/A). The data in **Figure 6e, f** was obtained from experiments with KRASG12D; fTg eIF2 α S/S mice and not with KRAS G12D mice containing endogenous/WT eIF2 α allele.

Question#7 (cont'd) Being this the case one wonders if then even the low-eIF2 α patients would benefit from ISRIB treatment. Have the authors ever compared the therapeutic effect of ISRIB when used in KRasG12D eIF2 α S/S vs KRasG12D eIF2 α A/A tumors? This would be important if eIF2 α expression levels are to be used as a prognostic marker as claimed by the authors. Similarly, could the authors compare the level of eIF2 α overexpression in tumors from KRasG12D ; fTg eIF2 α S/S and the KRasG12D exclusively harboring the endogenous eIF2 α alleles? How does this expression difference (fold change) compare to that observed in the clinic (high vs low eIF2 α patients)?

ANSWER: The most suitable approach to answer this question was the use of KRASG12D; fTg eIF2 α S/S, KRASG12D; fTg eIF2 α S/A and KRASG12D; fTg eIF2 α A/A mice for the induction of lung tumors and ISRIB treatments as described in **Fig. 6e, f**. The predicament with this approach is the time-consuming process to obtain the necessary number of mice and the prolonged time required for monitoring tumor development during ISRIB treatment (> 1 year; **Fig. 6e, f**). After

consulting with the Editor [REDACTED] on September 23, 2020, we proposed a modified approach, which was the subcutaneous transplantation of mouse KRAS G12D S/S and A/A cells in syngeneic mice followed by treatment with either vehicle control or ISRIB (10 mg/kg) after the formation of sizable tumors (see data below; arrowhead in the graph indicates the initiation of treatment of subcutaneous tumors of 200 mm³). The data from these experiments showed that KRAS G12D S/S cells produced large tumors, which responded to the anti-tumor effects of ISRIB. However, KRAS G12D A/A cells exhibited a substantially reduced growth, which hindered the treatments of mice with ISRIB (data below). This experiment provided further evidence for *the anti-tumor effects of ISRIB after the development of detectable lung tumors in mice*. The anti-tumor action of ISRIB in KRAS G12D tumors in subcutaneous transplantation assays in mice is included in **Supplementary Fig. 11a**.

Further to the point raised by the Reviewer, we would like to explain the data in **Fig. 6a, b** showing the anti-tumor effects of ISR inhibitors (PERK inhibitor and ISRIB) in mice transplanted with H1299 cells overexpressing KRAS G12C or WT KRAS. As included above in our response to Question #3, KRAS G12C overexpression in H1299 cells upregulates ISR signaling (i.e. increased p-PERK/p-eIF2) and renders the cells susceptible to ISR inhibition (i.e. colony formation assays) compared to KRAS WT-overexpressing cells despite the presence of mutant NRAS in the cells (**Supplementary Fig. 6, 7**). The increased susceptibility of the KRAS G12C-expressing H1299 cells to ISR inhibition in mice is in line with the tissue culture data and further supports the

interpretation that lung tumors with upregulated ISR are more susceptible to ISR inhibition than lung tumors with WT KRAS.

Question#7 (cont'd) *Similarly, could the authors compare the level of eIF2 α overexpression in tumors from KRASG12D; fTg eIF2 α S/S and the KRASG12D exclusively harboring the endogenous eIF2 α alleles? How does this expression difference (fold change) compare to that observed in the clinic (high vs low eIF2 α patients)?*

ANSWER: We have not used the KRAS G12D mouse exclusively harboring endogenous eIF2 because this mouse was not the proper control for p-eIF2 deficient (A/A) mouse. As explained above, the way the phospho-eIF2 knock in mouse (A/A) was developed required the comparison KRASG12D; fTg eIF2 α S/S and KRASG12D; fTg eIF2 α A/A lung tumors (Cell Metab. 2009;10(1):13-26). We inquired about the possibility of obtaining lung tumor sections from another lab using the KRAS G12D mouse model harboring endogenous WT eIF2 α alleles. After careful consideration, we have remained hesitant at this approach because of the different conditions used by us and them to induce lung tumor formation (i.e. CRE lentivirus in our experiments vs CRE adenovirus in their experiments, TP53 shRNA in our system vs TP53f/fl in their system). We reasoned that such different conditions are very likely to affect the timing of lung tumor progression and skew the interpretation of the results from the comparison of the two model systems.

Minor points:

a) The use of the eIF2 α S51A comes with no explanation whatsoever. I interpret this allele as a non-phosphorylatable and therefore a PERK insensitive inactive control, but it should be indicated for clarity.

ANSWER: This is correct. eIF2 α S51A defines a mutant form of eIF2 α that cannot be phosphorylated at S52. This is better explained in the revised version.

b) A similar work reporting the use of ISR inhibitors on lung adenocarcinoma cells in vitro and upon implantation has been recently reported (DOI: 10.1158/1541-7786.MCR-19-0245).

ANSWER: This nice study determined the role of ISR in LUAD cell motility in response to metabolic stress. The data support a positive role of ISR in tumorigenic responses elicited by metabolic stress in LUAD cells. This is now Ref. 50.

REVIEWERS' COMMENTS

Reviewer #1 (Remarks to the Author):

The authors have addressed all my questions/concerns in a comprehensive manner.

Reviewer #2 (Remarks to the Author):

The revised manuscript is much improved, all previous concerns have been addressed.

Reviewer #3 (Remarks to the Author):

The authors have now convincingly addressed all my concerns and in my opinion the manuscript is ready for publication